# Contractile forces in platelet aggregates under microfluidic shear gradients reflect platelet inhibition and bleeding risk

Lucas H. Ting[1], Shirin Feghhi[1], Nikita Taparia[1], Annie O. Smith[1], Ari Karchin[1], Esther Lim[2], Alex St. John[2], Xu Wang[2], Tessa Rue[3], Nathan J. White[1,2,4] & Nathan J. Sniadecki [1,4,5]

Platelets contract forcefully after their activation, contributing to the strength and stability of platelet aggregates and fibrin clots during blood coagulation. Viscoelastic approaches can be used to assess platelet-induced clot strengthening, but they require thrombin and fibrin generation and are unable to measure platelet forces directly. Here, we report a rapid, microfluidic approach for measuring the contractile force of platelet aggregates for the detection of platelet dysfunction. We find that platelet forces are significantly reduced when blood samples are treated with inhibitors of myosin, GPIb-IX-V, integrin $\alpha_{IIb}\beta_3$, P2Y$_{12}$, or thromboxane generation. Clinically, we find that platelet forces are measurably lower in cardiology patients taking aspirin. We also find that measuring platelet forces can identify Emergency Department trauma patients who subsequently require blood transfusions. Together, these findings indicate that microfluidic quantification of platelet forces may be a rapid and useful approach for monitoring both antiplatelet therapy and traumatic bleeding risk.

---

[1] Mechanical Engineering, University of Washington, Seattle, WA 98195, USA. [2] Department of Emergency Medicine, University of Washington, Seattle, WA 98195, USA. [3] Biostatistics, University of Washington, Seattle, WA 98195, USA. [4] Bioengineering, University of Washington, Seattle, WA 98195, USA. [5] Institute for Stem Cell and Regenerative Medicine, University of Washington, Seattle, WA 98195, USA. Correspondence and requests for materials should be addressed to N.J.W. (email: whiten4@uw.edu) or to N.J.S. (email: nsniadec@uw.edu)

Platelets are the primary mediators of arterial thrombosis, which is the leading cause of cardiovascular death and disability worldwide[1]. Platelets contribute to hemostasis by forming aggregates that staunch bleeding and initiate coagulation[2]. Due to the key role of platelets in thrombosis and hemostasis, antiplatelet therapies are used to treat myocardial infarction and ischemic stroke and platelet transfusions are used to manage traumatic bleeding[3–5]. There is growing interest in platelet function testing in cardiology and trauma, but these tests have not been widely adopted into clinical practice.

Platelet function is typically measured by measuring their adhesion or aggregation responses to agonists including thrombin, collagen, adenosine diphosphate (ADP), and arachidonic acid (AA)[6]. However, these approaches do not fully capture the complexity of platelets, which includes multiple activation pathways, intracellular signaling with calcium influx, exposure of surface integrins, and, finally, cytoskeletal reorganization and contraction. As a result, current adhesion and aggregation-based measurement modalities have provided limited benefit and are not used routinely in the management of thrombosis and hemostasis[7].

Platelet cytoskeletal contraction contributes to the strength and stability of both primary platelet aggregates and during consolidation of fibrin-rich blood clots[8–10]. When platelets bind to von Willebrand factor (VWF) and collagen, it triggers events that mobilize intracellular calcium, initiate shape change, and release ADP and thromboxane A2 (TxA$_2$), which activate nearby platelets to join the growing platelet-rich plug[11]. A nascent plug is thought to be a loose conglomerate of platelets, being held together by platelet-to-platelet and platelet-to-matrix adhesions[2]. Myosin-based forces acting through integrin receptors can strengthen platelet-matrix adhesions[12–14] and mediate the cohesion of platelets[10,15]. Compaction of a plug by platelet forces reduces its porosity, thereby increasing the concentration and retention of agonists like ADP and TxA$_2$[16–18].

Earlier approaches have measured platelet forces in plasma or whole blood during clot retraction[19–21]. However, these viscoelastic approaches are dependent upon generation of thrombin or fibrin, making it difficult to isolate the contribution of platelets independently from fibrin generation. More recently, microscale sensors have enabled the measurement of platelet forces at the single-cell level[14,22–26]. With microfluidic approaches, it has been possible to study platelet adhesion and aggregation under more clinically relevant flow conditions[12,27–32]. Using microscale sensors and microfluidics together would allow one to analyze platelet forces under flow in a manner that is akin to platelet-rich plug formation during early hemostasis.

Here, we present our development of an approach for measuring platelet forces using a microfluidic device that contains an array of microscale blocks and flexible posts (Fig. 1a). The surfaces of the microchannel, blocks, and posts are coated with VWF and collagen to support platelet adhesion. There is a local gradient in the shear rate at the block and post, which initiates the formation of a platelet-rich plug. The contractile force produced by the platelet-rich plug is measured by the deflection of a post towards the block. We find that platelet forces are dependent on the activity of myosin, engagement of glycoprotein Ib-IX-V (GPIb-IX-V) and integrin α$_{IIb}$β$_3$ with their ligands, and activation by ADP or TxA$_2$. We also find that platelet forces are reduced in cardiology patients who are taking aspirin and in trauma patients who are at risk of bleeding due to coagulopathy. Our results suggest that measuring platelet forces in this manner can quantify platelet responses to a wide range of activators and identify trauma patients likely to require hemostatic intervention.

## Results

**Formation of platelet aggregates using high shear gradients.** Platelets can be induced to attach and are activated when they pass through regions of high shear, causing them to adhere to exposed matrix and aggregate into platelet-rich plugs[28–30]. We developed a microfluidic approach that uses a region of high shear as a mechanism to induce the formation of platelet aggregates to measure their contractile strength. A blood sample was flowed through a microchannel containing rigid blocks and flexible posts (Fig. 1a). These structures were fabricated in polydimethylsiloxane (PDMS; Fig. 1b and Supplementary Figure 1) and collagen and VWF were adsorbed to support platelet adhesion and activation. From our computational fluid dynamics (CFD) simulations (Fig. 1c), the minimum and maximum shear gradients at the blocks were calculated to be $-2.15 \times 10^6\,\mathrm{s}^{-1}\,\mathrm{mm}^{-1}$ and $5.74 \times 10^6\,\mathrm{s}^{-1}\,\mathrm{mm}^{-1}$, respectively, indicating that platelets experience rapid acceleration and deceleration in shear. We designed the blocks and posts in a staggered configuration to minimize disturbances in the flow for the other sensors that were downstream (Fig. 1d).

When testing a blood sample, we observed that platelets accumulated on the downstream portion of the block and formed an aggregate that encapsulated the post (Fig. 1e). We observed that these aggregates were comprised entirely of P-selectin-positive platelets and without other blood cells attached (Supplementary Figure 2). Platelets that passed through the microchannel without attaching were measured by flow cytometry and were not found to be activated (Supplementary Figure 3). Moreover, we found that adsorption of collagen and VWF onto the surface of blocks and posts was essential for the formation of the aggregates because coatings of 2% Pluronic F-127 prevented platelet adhesion under flow (Supplementary Figure 4). We considered the aggregates to be reminiscent of platelet-rich plugs that have been observed in previous studies[10,18,28,33]. We did not notice significant differences in the shape or size of the aggregates that formed on the upstream sensors as compared to the ones downstream. We also noted that strings of platelets formed along the bottom of the microchannel, but they did not interfere with the aggregates that formed on the sensors.

**Platelet force generation accompanies platelet activation.** We observed the formation of aggregates using a combination of phase and fluorescence microscopy (Supplementary Movie 1). We noted that platelets aggregated initially at the corners of a block and were suspended in the flow (Fig. 2a). We observed that platelets in an aggregate translocated toward its central region, while at the same time, the post was observed to deflect toward the block. Since the deflection of the post was opposite to the fluid drag acting it, we attributed its movement to the contractile force generated by the platelets. To measure platelet forces, we tracked the deflection of the post over time (Fig. 2b). We found that platelet accumulation increased with wall shear rate, but platelet forces per aggregate size were similar for shear rates at and above $5000\,\mathrm{s}^{-1}$ (Supplementary Figure 5).

When we collected whole blood samples in heparin tubes and labeled platelets with fluo-3, we observed a rise in intracellular calcium in the aggregates (Fig. 2c and Supplementary Movie 2), which occurred before the onset of platelet forces (Fig. 2d). The calcium signal subsided after it reached its maximum, but platelet forces continued to increase steadily. Platelets that formed an aggregate when calcium concentration was maximal ($t = 45\,\mathrm{s}$) were observed to have filopodia that extended to neighboring platelets (Fig. 2e). The morphology of the platelets is reminiscent of the shape change that occurs when platelets are activated on collagen-coated surfaces[34]. These results indicate that the

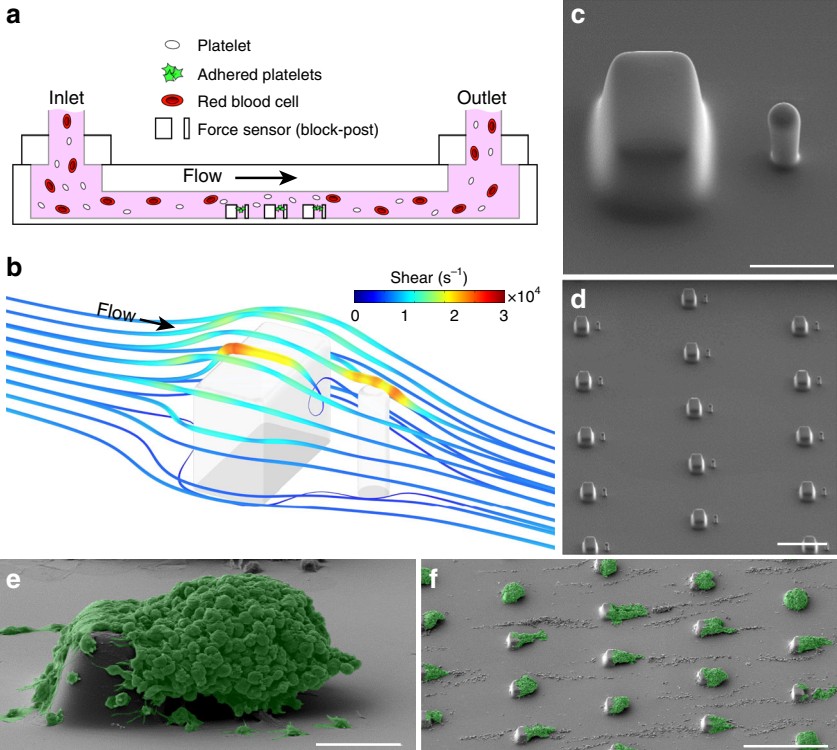

**Fig. 1** Microfluidic formation of platelet aggregates. **a** Schematic of microfluidic device in which whole blood is injected at the inlet and platelets aggregate onto arrays of microscale blocks and flexible posts for the measurement of platelet forces. **b** Computational fluid dynamics simulation at a wall shear rate of 8000 s$^{-1}$ show local regions of high shear that platelets encounter as they follow the streamlines that transit over a block and post. **c** Scanning electron microscopy (SEM) micrograph of a block and post at the bottom of the microchannel. Scale bar, 10 μm. **d** SEM micrograph of an array of blocks and posts. Scale bar, 50 μm. Pseudo-colored SEM micrograph of platelet aggregates that formed on (**e**) a block and post (scale bar, 10 μm) and **f** on an array of blocks and posts (scale bar, 100 μm) after 70 s of blood flow in the device

generation of platelet forces coincides with platelet activation upon adhesion to a block and post sensor.

Going forward, we collected blood samples in collection tubes containing sodium citrate, since these tubes are the standard for coagulation and platelet function testing. However, we were concerned that sequestration of extracellular calcium by sodium citrate may affect calcium signaling in the platelets and affect their force production. To address this concern, we compared blood samples collected in sodium citrate tubes, in lithium heparin, and blood samples drawn directly into an empty syringe without anti-coagulants (Supplementary Figure 6). We found that forces measured for samples collected in citrate, heparin, or without anti-coagulants were not statistically different.

**Platelet forces are sensitive to platelet inhibitors**. To examine the sensitivity of our measurement of platelet forces to inhibitors of myosin, adhesion receptors GPIb-IX-V and α$_{IIb}$β$_3$, and soluble agonists ADP and TxA$_2$, we tested blood samples from healthy donors that were divided into equal portions and either incubated with inhibitors or left untreated as a control. We inhibited non-muscle myosin using blebbistatin and observed that it strongly attenuated the generation of platelet forces (paired $T$-test $p = 0.049$) (Fig. 3a). We noted that blebbistatin produced platelet aggregates that were greater in size than the controls (paired $T$-test $p = 0.022$) (Fig. 3b). These results indicate the deflection of a post is due to myosin-generated forces and that myosin-inhibited platelets are unable to compact the aggregates, causing them to be larger is size.

Platelet aggregation is mediated in part by GPIb-IX-V and integrin α$_{IIb}$β$_3$[35,36], but these receptors can also transmit cytoskeletal tension[26,37]. We found that incubating with either AK2 to inhibit GPIb-IX-V or c7E3 to inhibit α$_{IIb}$β$_3$ led to a

difference in platelet forces (one-way analysis of variance (ANOVA) $F = 12.98$, DF = 2, $p = 0.001$) (Fig. 3c) and aggregate size (one-way ANOVA $F = 5.67$, DF = 2, $p = 0.0185$) (Fig. 3c, d). We noted that c7E3 had a greater inhibition of platelet forces (Tukey $p = 0.001$) and aggregate size (Tukey $p = 0.016$) than AK2 (Tukey $p = 0.019$ for force and $p = 0.11$ for area). In general, the shape of the aggregates that formed with AK2 were elongated, while those formed with c7E3 were much smaller in size (Supplementary Figure 7). These results indicate that our measurements require the function of GPIb-IX-V and α$_{IIb}$β$_3$ in order for aggregates to form on the block and contract the posts.

Activated platelets release TxA$_2$ and ADP, which act as positive feedback mechanisms for the growth of aggregates[36]. Antiplatelet therapies focus predominantly on one or both of these agonists to reduce the risk of arterial thrombosis. To examine whether our measurements of platelet forces are sensitive to these activation pathways, we tested blood samples with acetylsalicylic acid (ASA), which inhibits TxA$_2$ formation, or with 2-MeSAMP (2-methylthioadenosine 5'-monophosphate triethylammonium salt), which inhibits the P2Y$_{12}$ receptor for ADP. We found that ASA caused the aggregates to produce lower forces (Fig. 3e) and have smaller sizes as compared to controls (Fig. 3f). We also noted that the effect of ASA on platelet forces was dose dependent (Supplementary Figure 8). For ADP inhibition, we found that 2-MeSAMP reduced platelet forces (paired $T$-test $p = 0.003$) (Fig. 3g) and aggregate size (paired $T$-test $p = 0.0003$) (Fig. 3h). These result reveal that platelet forces are strongly influenced by TxA$_2$ and ADP signaling. We also note that percent force inhibition was greater than percent area inhibition for ASA and 2-MeSAMP, indicating increased inhibition of platelet forces compared to aggregation under high shear gradients.

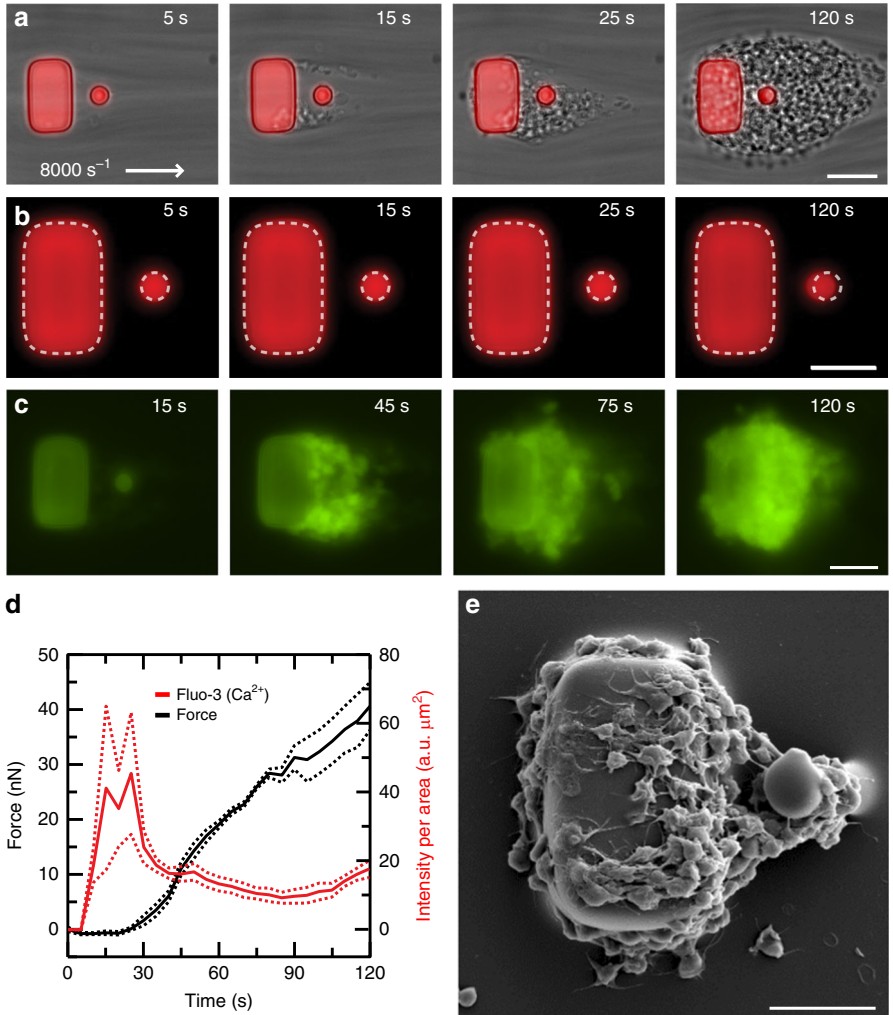

**Fig. 2** Measurement of platelet forces coincide with platelet activation. **a** Representative time course of phase contrast and fluorescence microscopy images taken during platelet aggregation at 8000 s$^{-1}$. Platelets clump initially at the trailing edges of the block and then connect to the post, eventually recruiting more passing platelets to attach to the aggregate. Scale bar, 15 μm. **b** The block and posts are fluorescently labeled (red, DiI), which allows platelet forces to be measured by analyzing the deflection of the post over time. Multiple aggregates are measured in the same field of view of the camera and averaged together over time to obtain a data trace. Scale bar, 10 μm. **c** Using a fluorescence indicator of Ca$^{2+}$ (fluo-3) shows that platelets exhibit increased intracellular calcium upon adhesion to the block. Scale bar, 10 μm. **d** Representative data from a single donor shows that Ca$^{2+}$ concentration (red) increases prior to platelet forces on the posts (black). Solid lines indicate the mean and dashed lines indicate standard error of the mean. **e** Scanning electron microscopy (SEM) micrograph of a platelet aggregate formed after 45 s at 8000 s$^{-1}$. Platelets attached to the block and post have undergone shear-induced activation and shape change. Scale bar, 10 μm

To examine if platelet forces were decreased in individuals taking antiplatelet medication, we collected blood samples from patients attending an outpatient cardiology clinic who were taking daily aspirin along with other medications (Supplementary Table 1). Similar to our in vitro experiments, we found these patients to have reduced platelet forces (T-test $p = 0.34$) (Fig. 3i) and aggregate size (T-test $p = 0.0017$) (Fig. 3j) as compared to healthy donors. These results with cardiology patients are congruent with our testing using ASA.

**Platelet forces predict need for blood transfusion in trauma.** Platelets contribute to hemostasis after trauma by providing clot strength and stiffness[38], and by adding resistance to fibrinolysis[39], both of which become impaired during trauma-induced coagulopathy (TIC), which is a mixed coagulopathy that manifests from consumptive coagulopathy, hyperfibrinolysis, and external influences such as hemodilution[40,41]. Evidence also suggests that platelet aggregation, and more so platelet contributions to clot

stiffness, can become impaired almost immediately after severe trauma, suggesting that intrinsic platelet dysfunction also plays an important role in the development of TIC[42–44]. Platelet function, when measured at Emergency Department arrival, is also strongly associated with mortality and blood transfusion requirements[42,45], and high ratios of platelet to packed red blood cell transfusions during trauma resuscitation are associated with decreased mortality[46]. Clinical measurement of platelet function has been typically measured by platelet count, which, when decreased, is related to mortality after trauma[47]. However, platelet count most often remains within a normal range early after trauma, making platelet count unable to guide early treatment decisions[47]. Given the evidence for acute platelet dysfunction as a potentially important, but poorly understood component of TIC and trauma mortality, we aimed to improve understanding of platelet dysfunction during TIC using an in vitro TIC model, in addition to examining the potential clinical utility of its early measurement for trauma medicine.

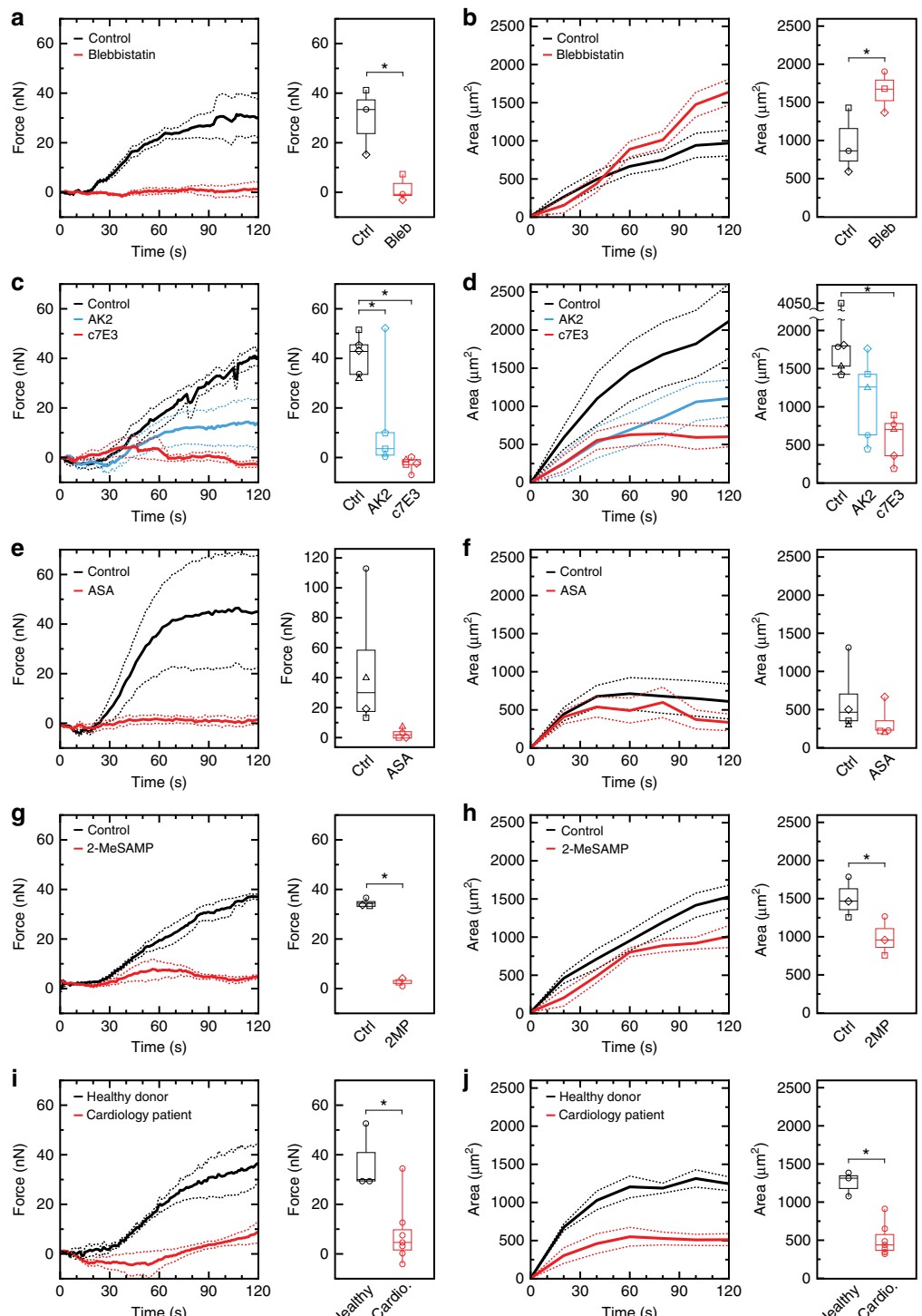

**Fig. 3** Platelet forces are sensitive to platelet inhibition. Forces and projected areas are shown for measurements of aggregates in the microfluidic device. Myosin inhibition (5 μM blebbistatin) causes **a** lower forces and **b** less compacted aggregates as compared to the control. Data are from three donors. Inhibition of GPIb-IX-V (5 μg mL$^{-1}$ antibody AK2) or integrin $\alpha_{IIb}\beta_3$ (20 μg mL$^{-1}$ antibody c7E3) causes a reduction in **c** platelet forces and **d** aggregate size. Data are from five donors. Inhibition of thromboxane formation using 0.3 mM acetylsalicylic acid (ASA) reduces **e** platelet forces and **f** aggregate size. Data are from four donors. Inhibition of adenosine diphosphate (ADP) activation (10 μM 2-MeSAMP (2-methylthioadenosine 5′-monophosphate triethylammonium salt)) reduces **g** platelet forces and **h** aggregate size. Data are from three donors. Cardiology patients on aspirin therapy have significantly reduced **i** platelet forces and **j** aggregate size as compared to healthy donors. Data shown are from seven cardiology patients and three healthy donors. Solid lines indicate the mean and dashed lines indicate standard error of the mean. Body of the box plots represent first and third quartiles. Center lines denote the median. Whiskers extend from the quartiles to the last data point within 1.5× interquartile range, with outliers beyond. Each marker shape in (**a**–**h**) represents control and inhibitor results for a blood sample from a single donor. Asterisks denote statistically significant differences (*$p < 0.05$) using paired two-way *T*-tests for (**a**, **b**, **e**–**h**), one-way analysis of variance (ANOVA) followed by Tukey's post-hoc test for (**c**, **d**), and two-way unpaired *T*-tests for (**i**, **j**)

For our measurement of platelet forces to be more reproducible and amenable for clinical use during acute bleeding situations like trauma, we developed a second-generation design of the microfluidic device and a bench-top system to directly measure platelet forces. The bench-top system contained a heated chamber that holds the microfluidic device at 37 °C, a syringe pump, and a custom-built microscopy system (Supplementary Figure 9). We increased the size of the block and post sensors in the second-generation design, but maintained a similar bending stiffness for the posts, so that its deflection would be larger when observed by the bench-top system (Supplementary Figure 10). We also increased the flow rate to $16,000\,s^{-1}$ in the first 15 s of testing so that a greater number of platelets attach and form aggregates that produced greater forces within 2 min of testing (Supplementary Figure 10). As a result of these changes, we were able to rapidly form platelet aggregates that produced twice the force as the first-generation design.

To improve our understanding of the role of platelet dysfunction during TIC, we first used a published in vitro model of TIC to examine for the presence of platelet contractile dysfunction, as suggested by several clinical studies (43–45). We used an in vitro model of TIC that reproduced a thromboelastography (TEG) response similar to that seen in trauma patients having TIC[48]. The published model reproduced TIC by 15% hemodilution, combined with clotting activation by tissue factor, and hyperfibrinolysis by tissue plasminogen activator. We first confirmed that the TIC model produced TEG curves similar to those seen in trauma patients with TIC (Supplementary Figure 11). We then found that platelet forces were significantly decreased with the TIC model as compared to controls (paired $T$-Test $p = 0.021$) (Supplementary Figure 11). To further investigate for an influence of hemodilution on platelet forces, we measured platelet forces in blood samples that were diluted with 0%, 15%, and 60% by volume with Ringer's solution and noted that forces were reduced with statistical significance (one-way ANOVA $F = 11.903$, DF = 2, $p = 0.0008$) at 15% dilution (Tukey $p < 0.01$) and 60% dilution (Tukey $p < 0.01$) (Supplementary Figure 12). These results indicate that platelet forces are sensitive to factors that contribute to TIC and may be potential useful as a clinical indicator of TIC.

We then examined if platelet forces were also reduced in vivo after trauma, and whether their measurement could be clinically useful. To answer these questions, we conducted a prospective cross-sectional observational study of Emergency Department trauma patients at a regional trauma center. Blood was sampled on arrival to the Emergency Department from 110 trauma patients and 10 healthy control subjects for comparison (Supplementary Table 2). We compared platelet forces in trauma patients who required blood product transfusion within the first 24 h of hospital care, a standard surrogate for severe bleeding, to trauma patients who did not require any transfusions, as well as to non-injured, healthy donors who served as control subjects (Fig. 4a). Seventeen (15%) patients received any blood transfusions within 24 h. Transfusions were given in a balanced, ratio-driven protocol, and hence all patients received a mix of blood products. Those transfused received an average (SD) of 3.3 (4.4) packed red blood cell units (79% within the first 4 h), 1.3 (1.5) platelet units (63% within 4 h), and 2.8 (4.9) plasma units (85% within 4 h).

We observed that mean (SD) platelet forces were different between groups (one-way ANOVA $F = 4.4$, DF = 2, $p = 0.015$), and were decreased for the transfused trauma group at 82.8 (40.5) nN when compared to healthy controls at 148.2 (44.4) nN (Tukey $p = 0.017$) and the non-transfused trauma group at 122.9 (60.1) nN (Tukey $p = 0.044$). We analyzed the receiver operating characteristics (ROC) for platelet forces and the need for any blood product transfusion within the first 24 h of hospitalization after trauma (Fig. 4b). Platelet forces significantly predicted the need for transfusion with an area under the ROC curve of 0.72 ($p = 0.006$). The unit odds ratio indicated that for every one nanonewton increase of force, there was a significant decrease of 1.4% in the odds of receiving a transfusion within 24 h (inset in Fig. 4b). Platelet count was in the normal range for the three groups (Fig. 4c) and there were no significant trends between platelet count and platelet forces (Supplementary Figure 13) or between fibrinogen concentration and platelet forces in the trauma patients (Supplementary Figure 14).

Given the effect of in vitro hemodilution on platelet forces, we also examined for potential confounding effects of hemodilution and prehospital (ambulance) administration of intravenous fluids on the predictive capability of platelet force measurements. Blood hematocrit was not significantly decreased in those patients requiring transfusion (Supplementary Table 2). Seventy-six percent of subjects who required transfusion received prehospital intravenous fluids, while 63% of non-transfused subjects received prehospital fluids (chi-square $p = 0.29$). The average volume of prehospital fluids administered was not different between these groups either (Supplementary Table 2). The mean (SD) platelet force was not significantly decreased in those receiving prehospital fluids at 106.1 (68.4) nN versus those who did not receive any prehospital fluids at 120.1 (58.4) nN ($T$-test $p = 0.29$). There were also no significant associations between platelet forces and hematocrit (Pearson $R = 0.5$, $p = 0.59$) or volume of prehospital fluids received (Pearson $R = -0.06$, $p = 0.61$). To further examine for possible confounding effects of hematocrit and prehospital fluid administration on the association between platelet forces and transfusion need, we performed a multivariate logistic regression model including these parameters. The multivariate model was significantly predictive of 24 h transfusion need (whole model chi-square $p = 0.002$), and platelet forces remained significantly and independently predictive of transfusion (platelet force effect likelihood ratio (LR) $p = 0.01$) after adjusting for hematocrit (hematocrit effect LR $p = 0.003$), and the presence of prehospital fluid administration (prehospital fluid effect LR $p = 0.3$). Therefore, we can conclude that there exists an important and independent role of platelet forces in trauma patients that is independent of hemodilution.

Rotational thromboelastometry (ROTEM) and impedance aggregometry (Multiplate) were also measured. We found that maximal clot firmness (MCF) by extrinsic pathway activation by low dose tissue factor (EXTEM) was decreased in whole blood (one-way ANOVA $F = 3.9$, DF = 2, $p = 0.02$) for the transfused trauma group vs. non-transfused trauma group (Tukey $p = 0.017$) (Fig. 4d), but was not different between any groups when measured in plasma (one-way ANOVA $F = 1.05$, DF = 2, $p = 0.35$) (Fig. 4e). This result suggests that there existed platelet-specific dysfunction in the transfused trauma groups that negatively impacted whole-blood clot formation. There were no differences in ROTEM clot formation time (CFT) between groups when measured in whole blood (one-way ANOVA $F = 0.77$, DF = 2, $p = 0.46$) (Fig. 4f) and in plasma (one-way ANOVA $F = 0.09$, DF = 2, $p = 0.91$) (Fig. 4g). EXTEM measurements were also found to be independent of platelet count ($R^2$ values for MCF in whole blood: 0.109, CFT in whole blood: 0.14). Platelet aggregation responses measured by the area under the aggregation curve (AUC) were significantly different when aggregation was activated by ADP (one-way ANOVA $F = 3.88$, DF = 2, $p = 0.023$) (Fig. 4h), ristocetin (one-way ANOVA $F = 3.23$, DF = 2, $p = 0.043$) (Fig. 4k), and AA (one-way ANOVA $F = 9.02$, DF = 2, $p < 0.001$) (Fig. 4l). There were no differences in aggregation between groups when activated by thrombin activating peptide (TRAP) (one-way ANOVA $F = 0.84$, DF = 2, $p = 0.43$) (Fig. 4i) or collagen (one-way ANOVA $F = 2.11$, DF = 2, $p = 0.13$)

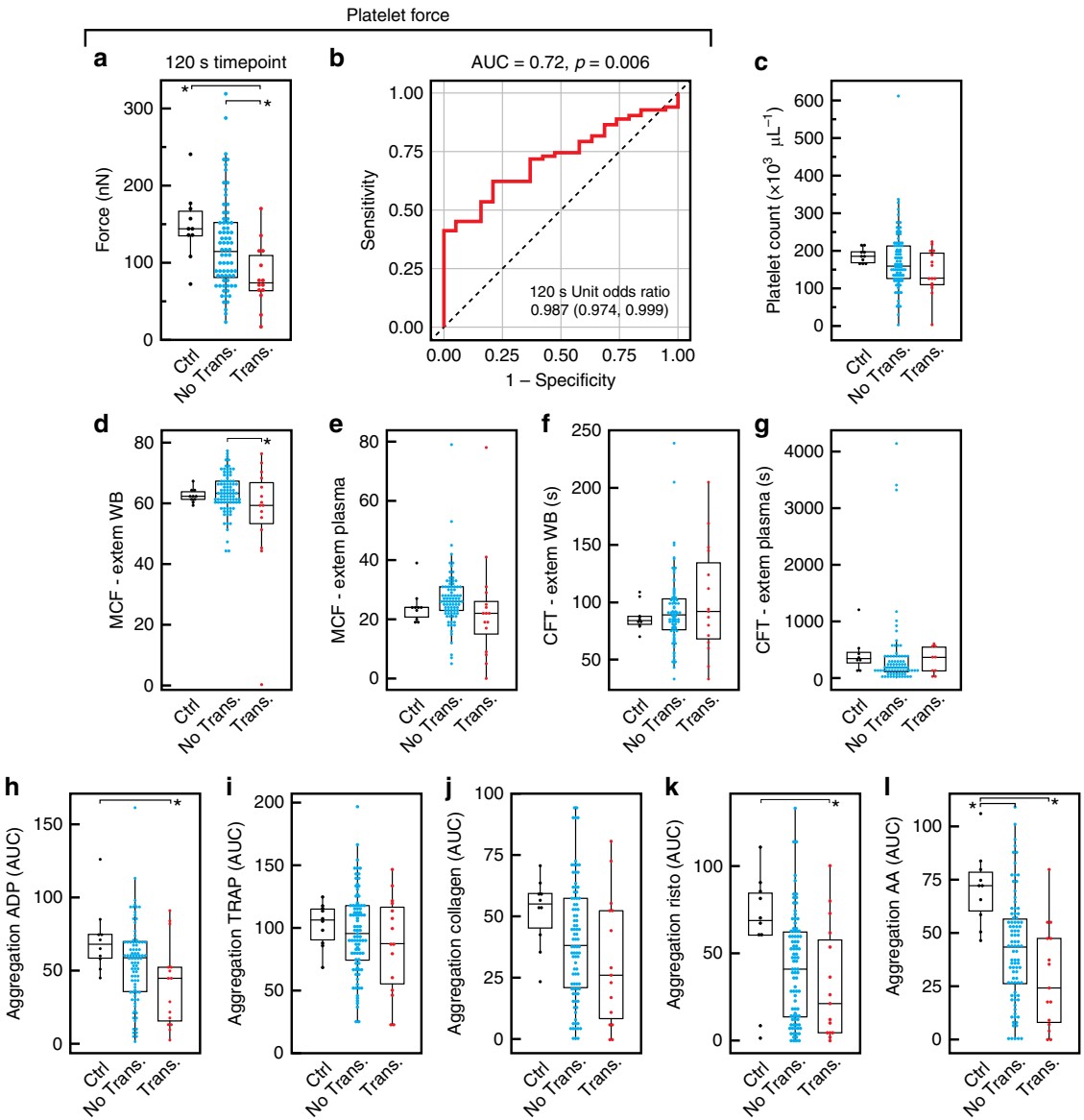

**Fig. 4** Platelet forces are indicative of bleeding risk in trauma patients. In a prospective cross-sectional observational study of Emergency Department trauma patients at a regional trauma center ($N = 110$ patients), blood was sampled upon arrival to the Emergency Department. **a** Platelet forces differed among healthy controls and trauma patients when stratified by need for blood product transfusion within the first 24 h of hospital care. Mean force was significantly decreased for those requiring transfusion (red diamonds) compared to both healthy controls (black diamonds) and vs. trauma subjects who did not require transfusion (magenta diamonds). **b** Receiver operating characteristics revealed that platelet forces significantly predicted the need for transfusion. **c** Platelet count was in the normal range for all three groups and was not different between groups. **d, e** Rotational thromboelastometry (ROTEM) showed that maximum clot firmness (MCF) was decreased in whole blood for transfused vs. not transfused trauma groups, but was not different between groups when measured in plasma. **f, g** There were no differences in ROTEM clot formation time (CFT) between groups when measured in whole blood and in plasma. **h–l** Platelet aggregation responses measured by the area under the aggregation curve (AUC) were significantly different when aggregation was activated by adenosine diphosphate (ADP), ristocetin, and arachidonic acid. There were no differences in aggregation between groups when activated by thrombin activating peptide (TRAP), or collagen. Platelet aggregation did not distinguish between transfused and non-transfused trauma patients. Body of the box plots represent first and third quartiles. Center lines denote the median. Whiskers extend from the quartiles to the last data point within 1.5× interquartile range, with outliers beyond. Data points represent results from a single subject. Asterisks denote statistically significant differences (*$p < 0.05$) using one-way analysis of variance (ANOVA) followed by Tukey's post-hoc test

(Fig. 4j). The transfused trauma group showed decreased aggregation vs. healthy controls only for ADP (Tukey $p = 0.018$) and ristocetin activation (Tukey $p = 0.036$). For AA activation, aggregation was decreased in both the non-transfused trauma group (Tukey $p = 0.002$) and the transfused trauma group (Tukey $p < 0.001$) when compared to healthy controls. There were no significant differences in aggregation for trauma patients between the transfused and non-transfused groups for any

activators (Supplementary Table 3). Aggregation AUC measurements were also independent of platelet count ($R^2$ values for ADP: 0.242, TRAP: 0.125, collagen: 0.113, ristocetin: 0.167, AA: 0.269). Taken together, platelet forces, ROTEM, and aggregometry data support an effect of trauma on platelet function. However, only platelet forces were able to distinguish transfused trauma patients conclusively from both healthy controls and non-transfused trauma patients.

## Discussion

We developed a rapid, microfluidic approach to measure the contractile forces of platelets in a blood sample. Maximal force generation requires platelets to have functional surface receptors, activation pathways, and contractile machinery. We found that measuring platelet forces could differentiate blood from healthy donors versus those on antiplatelet therapy and was predictive of the need for blood transfusion after traumatic injury. This work indicates that assaying platelet forces may provide a more integrative measurement of platelet function, combining aspects of adhesion, activation, and aggregation into a single assay that is capable of detecting platelet dysfunction arising from many causes.

A unique feature of our approach is the use of shear gradients to induce rapid platelet aggregation at our block and post sensors. The use of shear gradients in conjunction with collagen and VWF coatings to promote platelet adhesion does not require additional agonists or purification steps to separate platelets from a blood sample. The shear gradients we used were higher than those in previous microfluidic approaches, being twice as large as those used by Nesbitt et al.[32] and three orders of magnitude larger than Jain et al.[29], but they were designed to activate only the platelets near the block structures and not those in the bulk of the blood sample. As a result, we saw that platelets were activated once they attached to the sensors and not after they transited the length of the microchannel.

The magnitude of platelet forces we measured for shear-induced aggregates were similar to our prior measurements with micropost arrays, both being in the range of tens to hundreds of nanonewtons[22,49]. In comparison, we and others have conducted single platelet force assays and have found that individual platelets can produce tens of nanonewtons of force[23–26]. Given the high degree of contractility by a single platelet, it is likely that the complex structure of an aggregate provides a three-dimensional environment in which some of the platelets are not achieving their maximum contraction and/or the force produced by some of the platelets are captured by the deflection of the posts. In contrast to previous assays, our approach may be more amenable to point-of-care applications, since the readings can be collected rapidly and uses a bench-top system that does not require immunohistochemistry or confocal microscopy to obtain a measurement.

The prevention of acute arterial thrombosis focuses on antiplatelet therapy rather than anticoagulation. Arterial thromboses are typically described as white clots reflecting their predominant platelet-rich composition, whereas venous clots typically appear red due to red blood cells entrapped within the fibrin mesh[50]. However, goal-directed titration of antiplatelet medication guided by aggregometry has not yielded improved outcomes in cardiovascular disease[51]. In comparison, we demonstrate that measuring platelet forces could differentiate blood from normal and aspirin-treated individuals and, furthermore, could differentiate between different ASA doses (Supplementary Figure 8). We also find that our measurement of platelet forces is sensitive to inhibition of P2Y$_{12}$. Platelet forces may, therefore, provide the increased fidelity that is needed to guide antiplatelet therapy, but further studies are required.

There have been many attempts to rapidly predict the need for blood transfusion after trauma, mostly focusing on multivariate models incorporating vital signs, mechanism of injury, and bedside ultrasonography to predict the need for large volume transfusions[52]. Platelet count is commonly used to identify coagulopathy and guide blood product transfusions but, typically, these counts are well within the normal range, despite active bleeding[47]. Laboratory plasma clotting assays applied to trauma are useful to predict transfusion needs, but are generally hampered by slow processing and reporting times and lack of relevance to platelet transfusion[53]. Near-patient viscoelastic tests of clot formation are also being used to provide more holistic assessments of clot formation and to guide transfusion strategies during trauma resuscitation[54]. However, they are unable to detect most types of platelet inhibition in trauma patients, leaving a significant and important diagnostic gap, given the important association found between platelet dysfunction and mortality[55]. Rapid measurement of platelet force that is sensitive to both platelet inhibitors and intrinsic platelet dysfunction arising during TIC could potentially fill this gap. There are many possible factors that may contribute to platelet dysfunction after trauma, and further mechanistic investigation and prospective clinical trials are required to confirm the utility of these measurements. However, when combined with current clinical and laboratory-based testing algorithms, platelet force measurement has potential to become a useful addition to current methods used to rapidly identify bleeding risk and determine transfusion needs during acute bleeding situations.

## Methods

**Microfluidic device.** The microfluidic devices were fabricated in PDMS (Sylgard 184, Dow Corning) by soft lithography from a silicon master made using SU-8 2000 series negative photoresist to achieve high aspect ratio structures (Supplementary Figure 1). For experiments shown in Figs. 1–3, we used a master with blocks that were 15 μm × 15 μm × 25 μm and posts that were 4 μm in diameter and 15 μm in height. For experiments related to the trauma observational study, we used a second-generation design with blocks that were 25 μm × 25 μm × 15 μm and posts that were 6 μm in diameter and 25 μm in heights.

**Surface coatings.** Rat tail collagen type I (200 μg mL$^{-1}$, BD Bioscience) in 0.1 M acetic acid was incubated in the microchannels for 1 h. After rinsing with Tyrode's buffer (10 mM HEPES, 138 mM NaCl, 5.5 mM glucose, 12 mM NaHCO$_3$, 0.36 mM Na$_2$HPO$_4$, 2.9 mM KCl, 0.4 mM MgCl$_2$), purified human VWF (10 μg mL$^{-1}$, Haematologic Technologies) in Tyrode's buffer was incubated for 12 h. After rinsing, 1,1'-dioleyl-3,3,3',3'-tetramethylindocarbocyanine methanesulfonate (DiI, Life Technologies) at 5 μg mL$^{-1}$ was absorbed into the PDMS for 1 h, following by additional rinsing.

**Blood collection.** Whole blood was collected from consenting donors under a protocol approved by institutional review board (IRB) at the University of Washington (UW) that complies with ethical regulations. For experiments shown in Fig. 2, we collected blood into lithium heparin tubes (90 USP, 6.0 mL, BD Vacutainer). For all other experiments, we collected blood in sodium citrate tubes (0.109 M/3.2%, 2.7 mL, BD Vacutainer). For the study involving cardiology patients, four males and three female donors, aged 19–68 years, taking daily aspirin doses of at least 75 mg and other medications (Supplementary Table 1), were recruited from UW's Regional Heart Center in accordance with an IRB protocol.

**Computational fluid dynamics (CFD).** Simulations of fluid–structure interactions were performed in COMSOL Multiphysics. The simulations included of the entire width and height of the microchannel, 10 mm of its length, and a block and post structure placed 3 mm from the outlet. The flow rate was defined at the inlet and relative pressure of zero at the outlet. Mesh independence and convergence was checked by running simulations from 640,000 to 2,940,000 elements. The position of the block and post was varied along the width of the microchannel to test for flow variations due to the walls, but they were found to be negligible. Drag forces acting on the posts produced an insignificant deflection (<0.1 μm) at the highest shear rate (16,000 s$^{-1}$)

**Microfluidic testing protocol.** Blood was drawn into a disposable syringe (BD Scientific) and then loaded onto a syringe pump (Harvard Apparatus). For experiments shown in Figs. 1–3, the flow rate was set to generate a wall shear rate of 8000 s$^{-1}$ based upon calculations. For experiments shown in Fig. 4, the flow rate was set to 16,000 s$^{-1}$ for 15 s to form aggregates more quickly than the previous experiments and then reduced to 500 s$^{-1}$ to reduce the attachment of additional platelets.

**Electron microscopy.** A solution of 4% paraformaldehyde was injected into a device after blood testing. The top layer of the microfluidic device was removed with a razor blade, and then the device was rinsed in deionized (DI) water and dried using a critical point dryer (Structure Probe, Inc.). The devices were sputtered with a 50 Å coating of Au (Structure Probe, Inc.) and imaged at 5 kV in a scanning electron microscope (FEI Sirion).

**Force and area analysis.** Using a Nikon Eclipse Ti microscope with a 60× oil objective with a live-cell chamber maintained at 37 °C, microfluidic devices were imaged with a cooled CCD camera (Andor Clara) using phase and fluorescence illumination to track the size of the aggregates and deflection of the posts. We calculated the force acting on each post by measuring its deflection over time. Using Euler–Bernoulli beam theory, as we have done before to measure cellular forces with microposts[22,26], we

assumed the force ($F$) produced by platelets to be given by $F=(3\pi ED^4/64H^3)\delta$, where $E$ is the elasticity modulus of PDMS (3.8 MPa), $D$ is the diameter (4 or 6 μm), $H$ is the height (15 or 25 μm), and $\delta$ is the deflection of the post. Using a custom-written code in MATLAB, we identified each post in an image frame and calculated its deflection, and hence force. Additionally, the projected area of a platelet aggregate was measured by manual tracing using Nikon NIS-Elements analysis tools.

**Intracellular calcium**. Blood samples collected in heparin tubes were incubated with fluo-3 (10 μM, Life Technologies) for 30 min at 37 °C. To monitor the calcium and force response, images of the fluo-3 and DiI signals were taken in an alternating order at 0.2 Hz. Due to the excitation and emission spectrum of DiI, there is a faint signal of the block and post from bleed-through in fluorescence imaging. The background DiI signal remained constant in each frame and did not interfere with the analysis of intracellular calcium by fluo-3. To quantify the calcium signal, the average pixel intensity of fluo-3 was analyzed within a region of interest drawn to match the size of the platelet aggregate in each frame.

**Inhibitor studies**. Inhibition of myosin was achieved by incubating blood with 5 μM blebbistatin (Sigma Aldrich) for 10 min prior to testing. Antibodies AK2 (5 μg mL$^{-1}$ Abcam, Cat. No. ab61402) and c7E3 (20 μg mL$^{-1}$ ReoPro®, Janssen Biologics) were incubated for 25 min. ASA was incubated for 20 min. Inhibition of receptor P2Y$_{12}$ was accomplished by adding 10 μM 2-MeSAMP (Sigma Aldrich) and incubating for 25 min. For all experiments, control samples had an equivalent volume of deionized (DI) water or dimethyl sulfoxide added.

**Trauma observational study**. Human subjects were enrolled as part of a single-center cross-sectional observational study taking place at Harborview Medical Center Emergency Department, an American College of Surgeons-verified Level I Trauma Center. Delayed consent was obtained for study participation from all study subjects in accordance with an approved IRB protocol. Written consent was obtained subsequently from each subject or authorized representative during the hospital stay. Subjects were screened for enrollment on arrival to the Emergency Department and were enrolled if they were adult (>18 years old), not pregnant or incarcerated, presented directly from the place of injury, and had received no more than three units of transfused blood products prior to a blood draw for the study. An additional 10 healthy uninjured subjects were enrolled for comparison after informed consent. Blood samples were collected from a peripheral intravenous line placed by the clinical care team and then immediately transported to the laboratory for testing within 60 min of draw. A portion of each sample was centrifuged at 2.7 × g for 15 min, so platelet-poor plasma could be used to measure prothrombin time, activated partial thromboplastin time, and fibrinogen concentration (STart-4 hemostasis analyzer, Diagnostica Stago). Rotational thromboelastometry (ROTEM Delta, Instrumentation Laboratory) was used to obtain clot formation profiles for whole blood and plasma using extrinsic pathway activation (EXTEM). Platelet aggregometry was performed in whole blood using impedance aggregometer (Multiplate, Roche Diagnostics) with ADP, collagen, TRAP, AA, and ristocetin reagents. Relevant clinical, laboratory, and outcome parameters were abstracted from the medical record including: demographics and outcomes, vital signs, cell counts, coagulation tests, arterial blood gas parameters, and number of blood product units transfused during the first 24 h of hospitalization, which is the timeframe of increased mortality from hemorrhage (Supplementary Table 2).

**Statistical analysis**. The vast majority of platelet force measurements were normally distributed (Shapiro–Wilk Goodness of Fit for normal distribution $p > 0.05$), and hence parametric methods were used for statistical analysis. Two-sided paired $T$-tests were used for comparison between experimental and control groups, while one-way analysis of variance (ANOVA) with Tukey's post-hoc test was used to compare the results with three or more groups at 120 s of assay time. Force and area curves are shown as an average of at least three biological replicates. The dashed lines in the curves indicate the standard error of the mean. Two-sided unpaired $T$-tests were used for comparison between healthy donors and cardiology patients or transfused and non-transfused trauma groups, where appropriate. Nominal logistic regression was used to determine the ROC of platelet force to predict the need for blood transfusion in the first 24 h of hospital care and unit odds ratios were calculated. Pearson's correlations were used to test for linear relationships between two continuous variables. Chi-square was used to test for differences between categorical variables, and multivariate logistic regression with LR tests were used to examine for individual effects of hematocrit and the presence of prehospital fluid administration on the relationship between platelet force measurements and blood product transfusion.

**Reporting summary**. Further information on experimental design is available in the Nature Research Reporting Summary linked to this article.

## Code availability

Analysis code and software used to analyze micropost sensors and run hardware is available on github (https://github.com/lhting/Trauma_Study_GUI).

## Data availability

All relevant data are available from the corresponding author on reasonable request.

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

## Acknowledgements

Special thanks to Kimsey Platten and Diana Chien for technical assistance during the trauma observation study. The authors would also like to thank Kuan-Fu Chen, Michelle Shieh, Adriane Magena Fura, and Carly Morgan for their assistance. This work is funded through a Defense Advanced Research Projects Agency Young Faculty Award (N66001-11-1-4129), the Coulter Foundation Translational Research Award, grants from the Life Science Discovery Fund (LSDF-7434512), University of Washington CoMotion, the National Science Foundation (CMMI-1402673), and the National Institutes of Health (UL1TR000423, KL2TR000421, EB001650). Part of this work was conducted at the University of Washington's Washington Nanofabrication Facility, a member of the National Science Foundation's National Nanotechnology Coordinated Infrastructure.

## Author contributions

N.J.S. and N.J.W. conceived the idea of measuring platelet forces for bleeding management. N.J.S., N.J.W., L.H.T, S.F., N.T, A.O.S., and A.K designed experiments, engineered the technical approach to measure platelet forces, performed in vitro experiments, and/or analyzed results. E.L, A.S.J., and X.W conducted the trauma study. T.R. performed statistical analyses of the data from the trauma study. All authors contributed to the writing of the manuscript and approved its final form.

## Additional information

**Competing interests:** N.J.S., N.J.W., L.H.T, A.K., A.O.S., S.F. and N.T. are shareholders, consultants, and/or employees of Stasys Medical Corp., a company created to develop the platelet force technology for clinical use. Two US patents have been filed covering the technology: US 9,213,024 B2 (Microfluidic devices for measuring platelet coagulation and associated systems and methods) and US 9,140,684 B2 (Device to expose cells to fluid shear forces and associated systems and methods). E.L, A.S.J., X.W. and T.R. declare no competing interests.

