## [Peer Review File · Nature Communications]

Reviewers' comments:

Reviewer #1 (Remarks to the Author):

Ting et al. describe a microfluidic device in which the retraction force of platelet aggregates formed under high shear gradients is measured by the deflection of PDMS post. The authors show that this retraction force is sensitive to platelet function through a series of experiments with inhibitors along with samples from individuals taking antiplatelet drugs and trauma patients. While other have measured platelet retraction forces, this is the first report to my knowledge, that does so with platelet aggregates formed under flow. There is some evidence in a cohort of trauma patients is that this approach has advantages over existing clinical platelet function assays. There are some control experiments and clarifications the authors could make that would strengthen the authors conclusions. These suggestions are outlined in the comments below.

Major Comments:

1. I have concerns that platelets could be pre-activated prior to or during their transit through the channel. This is suggested by the fact that all platelets are P-selectin positive (Fig. S2), the shear stress ($8000 \text{ 1/s} * 3.8 \times 10^{-3} \text{ Pa s} = 30.4 \text{ Pa}$) exceeds that reported to cause shear induced platelet activation (Shankaran et al, Blood, 2003), and blood without an anticoagulant yields similar forces as with anticoagulant (Fig. S4). A useful control experiment would be to show that perfusion through the device itself in the absence of adhesive substrates (collagen/VWF) does not lead to platelet activation (as measured by activation markers like integrin activation, P-selectin, or phosphatidylserine exposure). Along the same lines, do aggregates form on the block and post in the absence of these proteins (collagen/VWF)?
2. Fig. 2C-D. It is unclear how the authors decouple an increase in calcium fluorescence due to activation (release of calcium into the cytoplasm from intracellular stores) and continued platelet accumulation. In other words, in Fig. 3b it is shown that platelets accumulate over ~90 seconds, which would add to the Fluo-3 signal in addition to changes signal in previously aggregated platelets.
3. Fig. S4. The appropriate comparison here would be between citrated and heparinized blood since these are the two anticoagulants used in the study.
4. The use of different device designs in the inhibitor/characterization studies and trauma patient studies makes for a difficult comparison between the data in Figures 1-3 and Figure 4; the size of the post/block are different, the wall shear rate and duration is different (8000 1/s for 2 min vs $16,000 \text{ 1/s}$ for 15 sec followed by 500 1/s for 1:45 min), and perhaps the temperature is different (room temperature vs 37 C, I cannot tell from the text). The rationale for each of these changes is unclear other than a statement about producing greater forces in 2 minutes. How do each of these changes individually affect platelet aggregation and forces?

5. Fig. S8 does not show “aggregates that produce greater forces within two minutes of testing” in the modified design. Rather it shows the shear rates along the block and post. Is this data available to the reader?
6. The conclusion that ‘platelet forces are sensitive to impaired coagulation’ is not supported by the methodology. Fig. S9 describes platelet forces in non-recalcified citrated whole blood where coagulation is inhibited (TF in citrated blood won’t initiate coagulation without divalent cations) and there is no evidence of fibrin formation. Consequently, is the reduction in force shown in Fig. S9a due to solely to hemodilution? What does hemodilution look like in the absence of TF and tPA?
7. The statistical tests used assume a normal distribution in the data. What evidence is there that the data is normal?
8. It would be preferable and in line with Nature Research Journals to show all the data points in scatter plots (or box and whisker plots for large n’s) rather than bar charts of means with SEM. This provides more transparency with respect to the data distribution.
9. The Discussion could be expanded a bit to include a direct comparison to other microscale/single platelet approaches to measuring platelet function. How do the shear gradients in this device compare to those used by others (references 27-29)? What are the advantages of measuring aggregate forces over single platelet forces (as previously reported by these authors and others)?

Minor Comments

1. Electron Microscopy-the devices were sputtered with what? Gold? How much?
2. What is the time between collection of blood from trauma patients and testing within the microfluidic device?
3. Please report the p-values from Fig. 4a in one of the Tables.

Reviewer #2 (Remarks to the Author):

The authors have developed an interesting new microfluidic device, that is designed to detect integrated platelet functional responses under the influence of defined shear gradients. This device aims to mimic various platelet responses that occur during a typical haemostatic response in vivo. This is achieved by engineering an array of stable PDMS blocks into a microchannel, with each block having a flexible post positioned immediately downstream. Perfused platelets adhere and aggregate to the collagen/vWf coated blocks/posts under precisely controlled wall shear rates. The forming

aggregates generate contractile force that are transmitted to the flexible posts, resulting in bending of the posts toward the stable blocks against the direction of blood flow. The bending extent of the flexible posts is quantified and used to represent platelet contractile force. The authors confirm that defective platelet responses caused by impaired GPIIb-IIIa activation, ADP release, TXA₂ generation, vWf-GPIb; fibrinogen/GPIIb-IIIa interactions, or GPIIb-IIIa mediated cytoskeletal contractility, results in reduced platelet aggregation and contractile force.

To evaluate the potential clinical application of platelet contractile force, the authors first demonstrated a reduced force generation for patients on aspirin. To assess if platelet contractile force is predictive of bleeding risk, the authors measured platelet contractile force, along with a list of other standard tests, for 93 trauma patients, and demonstrated that the obtained platelet contractile force, but not aggregometry and thromboelastography, can predict whether a trauma patient requires blood transfusion. The authors thus conclude that platelet contractile force may be a useful adjunct to currently available methods to identify bleeding risk, especially to determine transfusion needs.

The design of this device is new, innovative, and represents an important technical advance in enabling the rapid assessment of platelet contractile function. The generated platelet contractile force appears to be an accurate measurement of platelet dysfunction and a potentially useful test to predict whether trauma patients require blood transfusion. This is perhaps not surprising, as many previous studies using thromboelastography have confirmed that mechanically weak clots (which may be partly related to reduced platelet contractility) represents a severe bleeding risk requiring immediate therapeutic intervention.

A clear rationale for why trauma-induced coagulopathy was the disease state selected for study has not been provided. Blood product transfusion in trauma patients is primarily due to hyperfibrinolysis, consumptive coagulopathy/DIC and thrombocytopenia, and rather than platelet dysfunction. Since this 'force' assay detects platelet dysfunction (without the influence of thrombin and fibrin formation), it is unclear what aspect of platelet dysfunction might be manifested in transfused trauma patients? Is the platelet dysfunction (or platelet force) confounded by consumptive coagulopathy, such as reduced fibrinogen and or von Willebrand factor, and thrombocytopenia? It seems surprising that platelet count per se does not cause or correlate with platelet force changes in trauma patients (Supp Fig 10).

The design of this device is new, innovative, and represents an important technical advance in assessing platelet contractile function, enabling a rapid single assay to detect platelet dysfunction induced by many causes. The assay detects a diverse array of platelet responses including adhesion, activation, aggregation and contractility.

Major concerns:

1. Although the authors state that platelet dysfunction may be a contributor to TIC, this is a condition primarily of hyperfibrinolysis and a consumptive coagulopathy/DIC. Although thrombocytopenia may be a component of TIC, although uncommon in these patients (table S2), the authors found no correlation with thrombocytopenia and platelet force generation. In addition, this was only mentioned in the results section in page 12. Further background information is required to illustrate why platelet dysfunction is felt to be such an important contributor to traumatic bleeding in this clinical scenario.

2. It is an interesting finding that contractile force is impaired in the patients who subsequently went on to require transfusion, however overall numbers of patients needing transfusion were small (n=17) and it is not clear from the data (in table S2) how many patients received platelet concentrates vs FFP etc. It would be interesting to know if the platelet contractile force dysfunction corrected following transfusion.

3. The authors should highlight that the decision to transfuse trauma patient is based on a variety of clinical parameters, including mechanism/extent of injury, anticipated blood loss, presence of ongoing bleeding and laboratory indices (Hb, platelet count, PT/APTT, fibrinogen, TEG/ROTEM data). This information guides the transfusion strategy and allows it to be tailored to a particular patient. While this device may predict who is more likely to require transfusions, is it likely that the decision to transfuse a patient will be influenced by the output of this device? The potential exists that a patient may be inappropriately transfused a blood product independent of clinical parameters, because of impaired platelet contractile force. Inappropriate transfusion has been demonstrated in numerous studies to be associated with increased in-hospital mortality. These issues should be highlighted in the manuscript.

4. For Fig 1-3, the channel height is 30 μm , blocks are 15 μm in height, thus the adherent platelets experience 50% stenosis, and there is a 15 μm gap for platelet aggregate formation. Considering type 1 collagen is a potent stimulus of platelets, do the forming aggregates cause occlusion? Is the aggregate formation a dynamic process or stable? The height was increased to 50 μm in Fig 4 for trauma patients, to increase the contractile force' range (went up to 150 nN from 40 nN in Fig 3). Were the platelet aggregates much larger? What was the reason to change the dimension for trauma patients? Can the authors detect a similar force reduction using 30 μm tall channels for the same transfused trauma patients? Or are the taller channels more sensitive in detecting platelet dysfunction/contractile force reduction?

5. It is unclear why platelet adhesion and contractile force are only optimal when blood is perfused at $8000s^{-1}$, and no force (and aggregates) was detected at $2000s^{-1}$ after 2 min perfusion in 30 μm height channels (Supp Fig 3). This assay condition means that platelet adhesion is absolutely VWF-dependent, however VWF levels were not assessed in trauma patients.

6. The platelet contractile force measurement is rapid and informative in a single assay, thus seems superior to currently used standard tests such as EXTEM and coagulation which involve multiple assays. However, as shown in Supp Table 3, the most striking difference between non-transfused and transfused trauma patients is the dramatically prolonged EXTEM-clotting time. This suggests multifactorial defects in these severe trauma patients, particularly the level and function of clotting factors, loss or consumption of fibrinogen and VWF. Is the reduced platelet contractile force caused in part, by collective loss of fibrinogen, VWF and haemodilution, in addition to platelet dysfunction as demonstrated by aggregometry (Fig 4, and supp Table 3)?

7. Most of the presented data reached p value <0.05 , some were statistically insignificant (Fig 3c, e), raising queries about the reproducibility of the tests.

8. It would be more informative to plot platelet count vs 'force' for individual transfused trauma patients. The same for fibrinogen level, and EXTEM clotting time.

Minor points:

9. It may be more informative to present Fig. 4 bar graph on trauma patients in whisker box plot or dots overlays.

10. In Supp Fig 4, platelet force was also measure for non-anticoagulated blood. Does the generated thrombin lead to clot formation and what impact does this have on platelet contractile force measurements?

11. Page 13, the citations for Fig. S8 should be Fig. S9. Fig. S8 is a CFD characteristics figure.

12. In Fig 4 legend, MCF is defined as maximal clot force. This is incorrect and should be maximal clot firmness (and is correctly defined on page 14).

Reviewer #3 (Remarks to the Author):

The authors present data on a new method of assessing platelet aggregation and contractile force using microfluidics and microscale sensors. The block and sensor post are coated in collagen and vWF to allow adhesion and activation of platelets which then form an aggregate around the block and post, with subsequent contraction pulling the post towards the block. The contraction is myosin-dependent and sensitive to inhibitors of P2Y12 receptor activation and thromboxane A2 release as well as GPIIb/IIIa inhibition. Overall this is an interesting and imaginative way of assessing platelet function. They furthermore show a closer association between contractile force and transfusion requirement than is seen for any single-agonist-induced platelet aggregation response in trauma patients.

Comments:

1. The method seems to be much more sensitive than the PFA-100 analyser and this is a strength in assessing mild platelet function defects that may affect hemostasis. However, the marked sensitivity to aspirin, which seems to abolish the detected force of platelet contraction, means that this measure would not be able to discriminate between mild and more severe platelet function defects such as discriminating between patients on aspirin alone or dual antiplatelet therapy with aspirin and a P2Y12 inhibitor. Consequently this represents a limitation of the methodology compared to conventional platelet aggregometry. Further studies would be required to determine whether the measure of aggregate area could discriminate between single and dual antiplatelet therapy.
2. It is hard to assess claims of potential utility in phenotyping patient platelet function when the data are presented as mean and SEM. Dot plots would allow more scrutiny of how well the method discriminates between different platelet phenotypes compared to other methods. For example, light transmittance aggregometry using arachidonic acid as an agonist has excellent discrimination between control and aspirin-treated platelets and the presented method may not have this level of discrimination.
3. Since the presented method uses both collagen and vWF to activate platelets, it would have been interesting to look at how the results compared to platelet aggregation induced by a combination of collagen and ristocetin. The Multiplate results show lower responses to both collagen and ristocetin in the transfused patients, raising the question of whether combining the two would have achieved results closer to those obtained with the presented method. Perhaps post-hoc multiplication of the Multiplate collagen and ristocetin results might provide some insight here.
4. Figure 1: the labels for parts b and c are reversed.

August 17, 2018

Dear Reviewers:

In our revised manuscript, we have updated the data presented in Figures 2, 3, and 4, provided new data in the supplementary figures, and made modifications to the manuscript. Changes in the text are highlighted in red underlined font. These additions have helped to strengthen the quality of the manuscript. Below, we provide point-by-point responses to each of the reviewers.

Reviewer #1:

Ting et al. describe a microfluidic device in which the retraction force of platelet aggregates formed under high shear gradients is measured by the deflection of PDMS post. The authors show that this retraction force is sensitive to platelet function through a series of experiments with inhibitors along with samples from individuals taking antiplatelet drugs and trauma patients. While other have measured platelet retraction forces, this is the first report to my knowledge, that does so with platelet aggregates formed under flow. There is some evidence in a cohort of trauma patients is that this approach has advantages over existing clinical platelet function assays. There are some control experiments and clarifications the authors could make that would strengthen the authors conclusions. These suggestions are outlined in the comments below.

Major Comments:

1. I have concerns that platelets could be pre-activated prior to or during their transit through the channel. This is suggested by the fact that all platelets are P-selectin positive (Fig. S2), the shear stress ($8000 \text{ 1/s} * 3.8 \times 10^{-3} \text{ Pa s} = 30.4 \text{ Pa}$) exceeds that reported to cause shear induced platelet activation (Shankaran et al, Blood, 2003), and blood without an anticoagulant yields similar forces as with anticoagulant (Fig. S4). A useful control experiment would be to show that perfusion through the device itself in the absence of adhesive substrates (collagen/VWF) does not lead to platelet activation (as measured by activation markers like integrin activation, P-selectin, or phosphatidylserine exposure). Along the same lines, do aggregates form on the block and post in the absence of these proteins (collagen/VWF)?

We conducted new experiments with flow cytometry and find that platelets are not activated after their transit through the microfluidic device. We present this data in a new supplemental figure (Fig. S3) and have revised lines 198-199 in the manuscript. Specifically, we tested platelets in whole blood samples that were perfused through microfluidic devices coated with collagen and VWF or coated with Pluronics F-127, which blocks protein adhesion. We compared these sheared blood samples to unsheared blood (negative control) and unsheared blood that was activated with ADP (positive control). We find that platelets that were perfused through our device had levels of P-selection (CD62p) and

activated $\alpha_{\text{IIb}}\beta_3$ integrins (PAC-1) that were similar to platelets in unsheared blood, indicating that they were not activated by their transit through the microchannel.

2. Fig. 2C-D. It is unclear how the authors decouple an increase in calcium fluorescence due to activation (release of calcium into the cytoplasm from intracellular stores) and continued platelet accumulation. In other words, in Fig. 3b it is shown that platelets accumulate over ~90 seconds, which would add to the Fluo-3 signal in addition to changes signal in previously aggregated platelets.

The reviewer makes an excellent point. To address this issue, we revised our data analysis for calcium signaling in Fig. 2C. We now report the fluorescent intensity within a dynamic ROI that is defined by the projected area of the platelet aggregate in each time frame. We have revised lines 135-137 in the manuscript to clarify this analysis. We note that platelets accumulate on the surface of the aggregate over time, leading to a monotonic increase in size over time, but the calcium signal within the aggregate rises and then subsides indicating that calcium signaling is due to platelet activation rather than platelet accumulation.

3. Fig. S4. The appropriate comparison here would be between citrated and heparinized blood since these are the two anticoagulants used in the study.

We now show data for heparinized blood in Fig. S5 to compare three cases: no anticoagulant, citrated blood, and heparinized blood. Our data indicates that there is not a statistical difference in platelet forces for the three conditions. We have added this result to lines 230-232 in the manuscript.

4. The use of different device designs in the inhibitor/characterization studies and trauma patient studies makes for a difficult comparison between the data in Figures 1-3 and Figure 4; the size of the post/block are different, the wall shear rate and duration is different (8000 1/s for 2 min vs 16,000 1/s for 15 sec followed by 500 1/s for 1:45 min), and perhaps the temperature is different (room temperature vs 37 C, I cannot tell from the text). The rationale for each of these changes is unclear other than a statement about producing greater forces in 2 minutes. How do each of these changes individually affect platelet aggregation and forces?

We regret that we are unclear on the rationale for the design modifications between the first-generation and second-generation devices. This shortcoming was raised by another reviewer as well. To address this concern, we have added additional data and images in Fig. S9 to support our rationale and have provided a new paragraph about the design changes on lines 290-300.

Our first goal was to increase the visible deflection of the micropost by the platelets so that our bench-top system could track the movement of the posts with more precision. We accomplished this by increasing the height and diameter of the posts, while keeping their bending stiffness the same. We now show a comparison of post deflection for the two designs in Fig. S8 to emphasize this point. The height of the block was increased to match with the height of the post, but we rotated its position by 90° to improve its rigidity, *i.e.* by increasing its second moment of area.

Our second goal was to speed up the formation of aggregates on the sensors. We were guided by our data in Fig. S3, which indicates that increasing the shear rate (by increasing the flow rate) causes a faster accumulation of platelets on the sensors. As a result, we doubled the shear rate to 16,000 1/s for the first 15 seconds of testing so more platelets accumulated on the sensors and thereby imparted a greater force on the post. The shear rate was then decreased to 500 1/s to prevent new platelets from attaching to the sensors.

In both phases of the work, the temperature was maintained at 37 C, either through a live chamber box on the microscope or through a heater within the bench-top system.

5. Fig. S8 does not show “aggregates that produce greater forces within two minutes of testing” in the modified design. Rather is shows the shear rates along the block and post. Is this data available to the reader?

We provide force data and images of representative aggregates on the block and posts for the two designs of the microfluidic device in Fig. S9.

6. The conclusion that ‘platelet forces are sensitive to impaired coagulation’ is not supported by the methodology. Fig. S9 describes platelet forces in non-recalcified citrated whole blood where coagulation is inhibited (TF in citrated blood won’t initiate coagulation without divalent cations) and there is no evidence of fibrin formation. Consequently, is the reduction in force shown in Fig. S9a due to solely to hemodilution? What does hemodilution look like in the absence of TF and tPA?

We thank the reviewer for this insight. We conducted new experiments on hemodilution and find that platelet forces are significantly decreased at 60% dilution, but not at 15% dilution. We provide this new data in Fig. S11. We have corrected the concluding sentence on lines 311-313 to state: "These results indicate that platelet forces are sensitive to factors that contribute to TIC and may be potentially useful as a clinical indicator of TIC."

7. The statistical tests used assume a normal distribution in the data. What evidence is there that the data is normal?

For the inhibitor studies (Fig.3), their distributions were tested for normality using the Shapiro-Wilk Goodness of Fit Test after fitting to a continuous normal distribution. Only 4 of 22 distributions (18%) were found to be significantly skewed from the normal distribution (Shapiro-Wilk $p < 0.05$). There were no cases where comparisons were made between two conditions that were both significantly skewed. Consequently, we found it reasonable to use parametric methods for statistical analyses for this group of experiments.

For the trauma studies (Fig. 4), platelet forces for the trauma transfused and control groups were normally distributed (Shapiro-Wilk Goodness of Fit Test $p > 0.28$). The non-transfused trauma group was not normally distributed; however, the skewedness was due to only a single outlier. Therefore, we considered it reasonable to use parametric statistical tests to examine for differences in platelet force between these groups.

Subsequently, we have added a new sentence on line 169-171 for *Statistical Analysis* in the 'Materials and Methods' section: "The vast majority of platelet force measurements were normally-distributed (Shapiro-Wilk Goodness of Fit for normal distribution $p > 0.05$), so parametric methods were used for statistical analysis."

8. It would preferable and in line with Nature Research Journals to show the all the data points in scatter plots (or box and whisker plots for large n's) rather than bar charts of means with SEM. This provides more transparency with respect to the data distribution.

This concern was raised by the other reviewers as well. We have revised our graphs to be box-and-whisker plots with data points instead of bar graphs.

9. The Discussion could be expanded a bit include a direct comparison to other microscale/single platelet approaches to measuring platelet function. How does the shear gradients in this device compare to those used by others (references 27-29)? What are the advantages of measuring aggregate forces over single platelet forces (as previously reported by these authors and other)?

We thank the reviewer for this suggestion and have added two new paragraphs in the Discussion on lines 365-384 to compare our shear gradients and plate forces with prior methods:

"A unique feature of our approach is the use of shear gradients to induce rapid platelet aggregation at our block and post sensors. The use of shear gradients in conjunction with collagen and VWF coatings to promote platelet adhesion does not require additional agonists or purification steps to separate platelets from a blood sample. The shear gradients we used were higher than those in previous microfluidic approaches, being twice as large those used by Nesbitt et al. [32] and three orders of magnitude larger than Jian et al. [29], but they were designed to activate only the platelets near the block structures and not those in the bulk of the blood sample. As a result, we saw that platelets were activated once they attached to the sensors and not after they transited the length of the microchannel.

The magnitude of platelet forces we measured for shear-induced aggregates were similar to our prior measurements with micropost arrays, both being in the range of tens to hundreds of nanonewtons [22, 49]. In comparison, we and others have conducted single platelet force assays and have found that individual platelets can produce tens of nanonewtons of force [23-26]. Given the high degree of contractility by a single platelet, it is likely that the complex structure of an aggregate provides a three-dimensional environment in which some of the platelets are not achieving their maximum contraction and/or the forces produced by some of the platelets are captured by the deflection of the posts. In contrast to previous assays, our approach may be more amenable to point-of-care applications, since the readings can be collected rapidly and uses a bench-top system that does not require immunohistochemistry or confocal microscopy to obtain a measurement.

Minor Comments

1. Electron Microscopy-the devices were sputtered with what? Gold? How much?

The samples were sputtered with 50 Å of gold coating. We have revised the manuscript on line 115 to indicate the thickness of the coating.

2. What is the time between collection of blood from trauma patients and testing within the microfluidic device?

The blood was tested within 60 minutes from draw. We added this information to line 156 in our revised manuscript.

3. Please report the p-values from Fig. 4a in one of the Tables.

We now provide the p-values for platelet forces for the transfused and non-transfused trauma patients in the legend for Figure 4.

Reviewer 2:

The authors have developed an interesting new microfluidic device, that is designed to detect integrated platelet functional responses under the influence of defined shear gradients. This device aims to mimic various platelet responses that occur during a typical haemostatic response in vivo. This is achieved by engineering an array of stable PDMS blocks into a microchannel, with each block having a flexible post positioned immediate downstream. Perfused platelets adhere and aggregate to the collagen/vWf coated blocks/posts under precisely controlled wall shear rates. The forming aggregates generate contractile force that are transmitted to the flexible posts, resulting in bending of the posts toward the stable blocks against the direction of blood flow. The bending extent of the flexible posts is quantified and used to represent platelet contractile force. The authors confirm that defective platelet responses caused by impaired GPIIb-IIIa activation, ADP release, TXA2 generation, vWf-GPIIb; fibrinogen/GPIIb-IIIa interactions, or GPIIb-IIIa mediated cytoskeletal contractility, results in reduced platelet aggregation and contractile force.

To evaluate the potential clinical application of platelet contractile force, the authors first demonstrated a reduced force generation for patients on aspirin. To assess if platelet contractile force is predictive of bleeding risk, the authors measured platelet contractile force, along with a list of other standard tests, for 93 trauma patients, and demonstrated that the obtained platelet contractile force, but not aggregometry and thromboelastography, can predict whether a trauma patient requires blood transfusion. The authors thus conclude that platelet contractile force may be a useful adjunct to currently available methods to identify bleeding risk, especially to determine transfusion needs.

The design of this device is new, innovative, and represents an important technical advance in enabling the rapid assessment of platelet contractile function. The generated platelet contractile force appears to be an accurate measurement of platelet dysfunction and a potentially useful test to predict whether trauma patients require blood transfusion. This is perhaps not surprising, as many previous studies using thromboelastography have confirmed that mechanically weak clots

(which may be partly related to reduced platelet contractility) represents a severe bleeding risk requiring immediate therapeutic intervention.

A clear rationale for why trauma-induced coagulopathy was the disease state selected for study has not been provided. Blood product transfusion in trauma patients is primarily due to hyperfibrinolysis, consumptive coagulopathy/DIC and thrombocytopenia, and rather than platelet dysfunction. Since this 'force' assay detects platelet dysfunction (without the influence of thrombin and fibrin formation), it is unclear what aspect of platelet dysfunction might be manifested in transfused trauma patients? Is the platelet dysfunction (or platelet force) confounded by consumptive coagulopathy, such as reduced fibrinogen and or von Willebrand factor, and thrombocytopenia? It seems surprising that platelet count per se does not cause or correlate with platelet force changes in trauma patients (Supp Fig 10).

The design of this device is new, innovative, and represents an important technical advance in assessing platelet contractile function, enabling a rapid single assay to detect platelet dysfunction induced by many causes. The assay detects a diverse array of platelet responses including adhesion, activation, aggregation and contractility.

Major concerns:

1. Although the authors state that platelet dysfunction may be a contributor to TIC, this is a condition primarily of hyperfibrinolysis and a consumptive coagulopathy/DIC. Although thrombocytopenia may be a component of TIC, although uncommon in these patients (table S2), the authors found no correlation with thrombocytopenia and platelet force generation. In addition, this was only mentioned in the results section in page 12. Further background information is required to illustrate why platelet dysfunction is felt to be such an important contributor to traumatic bleeding in this clinical scenario.

To address the reviewer's concern, the following background information regarding the importance of platelet function to outcomes after trauma has been added to lines 278-289 to introduction the rationale for the studies on TIC and bleeding risk in trauma patients:

"Platelets are important contributors to hemostasis after trauma by contributing to clot strength [38] and regulating clot resistance to fibrinolysis [39], both of which are critical to the development of trauma-induced coagulopathy (TIC) [40, 41]. Clinical measurement of platelet function is typically measured by platelet count, which, when decreased, is related to mortality after trauma [42]. However, platelet count most often remains within a normal range early after trauma [42]. Alternatively, platelet function, measured by aggregation and platelet-contribution to clot formation is often decreased significantly almost immediately after severe trauma [43-45]. Platelet function measured at Emergency Department arrival is also strongly associated with mortality and blood transfusion requirements [43, 46], and high ratios of platelet to packed red blood cell transfusions during trauma resuscitation may be associated with decreased mortality [47]. For this reason, we investigated whether our measurement of platelet forces would be indicative of TIC and bleeding risk in trauma."

2. It is an interesting finding that contractile force is impaired in the patients who subsequently went on to require transfusion, however overall numbers of patients needing transfusion were small (n=17) and it is not clear from the data (in table S2) how many patients received platelet concentrates vs FFP etc. It would be interesting to know if the platelet contractile force dysfunction corrected following transfusion.

The reviewer makes an excellent point about post-transfusion measurements. We sought only to conduct a cross-sectional study examining platelet function upon arrival to the Emergency Department. Unfortunately, post-transfusion measurements were not part of our study design. We anticipate conducting a follow-on study to ascertain whether platelet forces are corrected following transfusion.

Regarding the number of patients who received transfusions, the following details regarding blood products has been added to lines 321-325:

"Seventeen (18%) of patients received any blood transfusions within 24 hours. Transfusions were given in a balanced, ratio-driven protocol, so all patients received a mix of blood products. Those transfused received an average (SD) of 3.3 (4.4) packed red blood cell units (79% within the first 4 hours), 1.3 (1.5) platelet units (63% within 4 hours), and 2.8 (4.9) plasma units (85% within 4 hours)."

3. The authors should highlight that the decision to transfuse trauma patient is based on a variety of clinical parameters, including mechanism/extent of injury, anticipated blood loss, presence of ongoing bleeding and laboratory indices (Hb, platelet count, PT/APTT, fibrinogen, TEG/ROTEM data). This information guides the transfusion strategy and allows it to be tailored to a particular patient. While this device may predict who is more likely to require transfusions, is it likely that the decision to transfuse a patient will be influenced by the output of this device? The potential exists that a patient may be inappropriately transfused a blood product independent of clinical parameters, because of impaired platelet contractile force. Inappropriate transfusion has been demonstrated in numerous studies to be associated with increased in-hospital mortality. These issues should be highlighted in the manuscript.

The reviewer highlights an important point regarding the potential clinical applications of platelet force measurement in relation to current approaches to blood product transfusion after trauma. We have revised lines 410-414 in the final paragraph in the 'Discussion' to emphasize that platelet force measurements should ultimately be used in conjunction with current clinical and laboratory parameters (ROTEM) to inform blood transfusion decisions:

"Rapid measurement of platelet force that is sensitive to both platelet inhibitors and to intrinsic platelet dysfunction arising during TIC would fill this gap. When combined with current clinical and laboratory-based testing algorithms, platelet force measurement has potential to become a useful addition to current methods used to identify bleeding risk and determine transfusion needs during acute bleeding situations."

4. For Fig 1-3, the channel height is 30 μm , blocks are 15 μm in height, thus the adherent platelets experience 50% stenosis, and there is a 15 μm gap for platelet aggregate formation. Considering type 1 collagen is a potent stimulus of platelets, do the forming aggregates cause occlusion? Is the aggregate formation a dynamic process or stable? The height was increased to 50 μm in Fig 4 for trauma patients, to increase the contractile force' range (went up to 150 nN from 40 nN in Fig 3). Were the platelet aggregates much larger? What was the reason to change the dimension for trauma patients? Can the authors detect a similar force reduction using 30 μm tall channels for the same transfused trauma patients? Or are the taller channels more sensitive in detecting platelet dysfunction/contractile force reduction?

The reviewer raises excellent questions regarding fluid mechanics, platelet accumulation, and occlusion. We would like to point out that questions on the differences between the first-generation (Gen 1) and second-generation (Gen 2) devices was raised by the previous reviewer. As stated earlier, we increased the size of the block and post, but maintained a similar bending stiffness to increase the deflection of the posts for our measurement with the bench-top system. We did not test blood samples from trauma patients with the 30 μm tall microchannels (Gen 1) because we modified our design to work with the benchtop system, which needed to be a stand-alone system that could be installed near the Emergency Department for expedited testing of blood samples.

Occasionally, platelet aggregates can occlude a microchannel if the test runs longer than 2 minutes. We can mitigate the risk of an occlusion by reducing the flow rate after forming aggregates and thereby reducing the rate of platelet accumulation. Aggregate formation is a dynamic process for the size of an aggregate grows steady as platelets pass over it at high shear rates. We find that once the flow is decreased to 500 s^{-1} or less, the accumulation of platelets is minimal.

5. It is unclear why platelet adhesion and contractile force are only optimal when blood is perfused at 8000 s^{-1} , and no force (and aggregates) was detected at 2000 s^{-1} after 2 min perfusion in 30 μm height channels (Supp Fig 3). This assay condition means that platelet adhesion is absolutely VWF-dependent, however VWF levels were not assessed in trauma patients.

We have revised lines 213-215 and the legend for Fig. S4 to convey that shear rates $>5000 \text{ s}^{-1}$ affect the rate of platelet accumulation, but the platelet force per aggregate size is similar. Aggregates do form at 2000 s^{-1} but they are small and do not produce enough force to visibly bend the post.

Platelet adhesion under high shear is highly dependent on VWF (Ref. 28 & 30). We find that inhibiting the attachment of platelets to VWF with antibody AK2 causes a distinct change in the size and shape of the aggregates (Fig. 3D and S6). We postulate that VWF levels in the trauma patients would affect the ability of their platelets to adhere to the block and post sensors, but we did not have the ability to measure VWF in these patients.

6. The platelet contractile force measurement is rapid and informative in a single assay, thus seems superior to currently used standard tests such as EXTEM and coagulation which involve multiple assays. However, as shown in Supp Table 3, the most striking difference between non-transfused and transfused trauma patients is the dramatically prolonged EXTEM-clotting time. This suggests multifactorial defects in these severe trauma patients, particularly the level and function of clotting factors, loss or consumption of fibrinogen and VWF. Is the reduced platelet

contractile force caused in part, by collective loss of fibrinogen, VWF and haemodilution, in addition to platelet dysfunction as demonstrated by aggregometry (Fig 4, and supp Table 3)?

We provide new analysis of platelet forces vs. fibrinogen concentration and find that there is not a correlation (Fig. S13). As stated earlier, we do not have the ability to conduct VWF panels, so we cannot provide an evaluation on its influence. We would like to point out that the effect of hemodilution was raised by another reviewer and we provide new data with healthy donors that indicates that dilution can affect platelet forces.

7. Most of the presented data reached p value <0.05, some were statistically insignificant (Fig 3c, e.), raising queries about the reproducibility of the tests.

We appreciate the reviewer's concern about the statistics for our data. We would like to point out that the results in Fig. 3c are statistically significant, but those in Fig. 3e are not. We now show the results for ASA in Fig. 3e as box-and-whisker plots with data points for each individual donor. The large variability between donors in the control condition for these experiments undermined our statistical test, resulting in p-values of 0.146 for force and 0.378 for area (two-tailed *t*-test). However, for each of the five donors tested, the addition of 0.3 mM ASA caused a reduction in platelet forces, so the trend was consistent. We also see a consistent trend in force reduction by ASA in Supplemental Figure S7.

8. It would be more informative to plot platelet count vs ‘force’ for individual transfused trauma patients. The same for fibrinogen level, and EXTEM clotting time.

We now provide color-coded data for platelet count and fibrinogen concentration versus platelet force in Figures S12 and S13, respectively, for the transfused (*red*) and non-transfused (*blue*) trauma patients. Due to our IRB restrictions, we are not able to compare EXTEM clotting time versus platelet force with individual data points.

Minor points:

9. It may be more informative to present Fig. 4 bar graph on trauma patients in whisker box plot or dots overlays.

This concern was raised by the other reviewers as well. We have revised our graphs in Figure 4, as well as Figure 3 and the Supplemental Figures, to be box-and-whisker plots with data points.

10. In Supp Fig 4, platelet force was also measure for non-anticoagulated blood. Does the generated thrombin lead to clot formation and what impact does this have on platelet contractile force measurements?

Due to the high shear environment, we did not observe thrombin activity leading to fibrin formation in our microfluidic device. Our flow conditions are more akin to arterial flows, so the platelet

aggregates we observe are like arterial thromboses, a/k/a 'white' clots. We did collect blood samples in lithium heparin, but did not observe significant differences in platelet forces as compared to non-anticoagulated blood (Fig. S4).

11. Page 13, the citations for Fig. S8 should be Fig. S9. Fig. S8 is a CFD characteristics figure.

Thank you, we have made this correction.

12. In Fig 4 legend, MCF is defined as maximal clot force. This is incorrect and should be maximal clot firmness (and is correctly defined on page 14).

We have corrected this mistake in the legend for Figure 4.

Reviewer 3:

The authors present data on a new method of assessing platelet aggregation and contractile force using microfluidics and microscale sensors. The block and sensor post are coated in collagen and vWF to allow adhesion and activation of platelets which then form an aggregate around the block and post, with subsequent contraction pulling the post towards the block. The contraction is myosin-dependent and sensitive to inhibitors of P2Y₁₂ receptor activation and thromboxane A₂ release as well as GPIIb/IIIa inhibition. Overall this is an interesting and imaginative way of assessing platelet function. They furthermore show a closer association between contractile force and transfusion requirement than is seen for any single-agonist-induced platelet aggregation response in trauma patients.

Comments:

1. The method seems to be much more sensitive than the PFA-100 analyser and this is a strength in assessing mild platelet function defects that may affect hemostasis. However, the marked sensitivity to aspirin, which seems to abolish the detected force of platelet contraction, means that this measure would not be able to discriminate between mild and more severe platelet function defects such as discriminating between patients on aspirin alone or dual antiplatelet therapy with aspirin and a P2Y₁₂ inhibitor. Consequently this represents a limitation of the methodology compared to conventional platelet aggregometry.

We agree that further studies would be required to determine whether the measurement of platelet forces could discriminate between antiplatelet therapy with aspirin alone or with aspirin and a P2Y₁₂ inhibitor. We would like to point out that our studies with 2-MeSAMP indicate that platelet forces are affected by P2Y₁₂ inhibition. However, we did not study aspirin and P2Y₁₂ inhibition together, so we cannot ascertain if this is a limitation of the methodology. We have modified lines 392-394 to highlight the results with P2Y₁₂ inhibition and state that further studies are required.

2. It is hard to assess claims of potential utility in phenotyping patient platelet function when the data are presented as mean and SEM. Dot plots would allow more scrutiny of how well the method discriminates between different platelet phenotypes compared to other methods. For example, light transmittance aggregometry using arachidonic acid as an agonist has excellent discrimination between control and aspirin-treated platelets and the presented method may not have this level of discrimination.

This concern was raised by the other reviewer. We now report our data as dot plots and/or box-and-whisker plots instead of bar graphs. By inspecting the result per individual, one can see that the inhibitors we studied has an effect on platelet forces that was nearly consistent. We do not that all but one of our cardiology patients taking aspirins had platelet forces that were lower than the forces measured for the healthy control donors.

3. Since the presented method uses both collagen and vWF to activate platelets, it would have been interesting to look at how the results compared to platelet aggregation induced by a combination of collagen and ristocetin. The Multiplate results show lower responses to both collagen and ristocetin in the transfused patients, raising the question of whether combining the two would have achieved results closer to those obtained with the presented method.

We appreciate this insight from the reviewer. We conducted a post-hoc multiplication of the results for collagen and ristocetin from Multiplate. Unfortunately, our results did not provide any addition insight as compared to the results individually. Specifically, a comparison of the results combining collagen and ristocetin has a p-value of 0.04 for control vs. transfused patients, but they were not statistically significant for control vs. non-transfused patients (p-value = 0.08) or non-transfused vs. transfused patients (p-value = 0.58).

4. Figure 1: the labels for parts b and c are reversed.

Thank you, we have corrected this mistake in Fig. 1.

To bring to the attention of all reviewers that we wrote the beam equation for the post incorrectly. We have corrected the equation in the revised manuscript on lines 123-126 to be consistent with our prior work with microposts and to properly disclose how the force was analyzed in this study.

In closing, we hope that the improvements in our manuscript address your concerns. On behalf of my fellow co-authors, we appreciated the valuable time of the reviewers and careful attention they have put into this process.

Sincerely,

Nathan J. Sniadecki, Ph.D.

Reviewers' comments:

Reviewer #1 (Remarks to the Author):

The authors have adequately addressed most of major comments in the revision and included several new experiments. There was one question from my previous review that was unanswered:

1. Fig. 3 addresses my previous comment regarding platelet activation caused by the device/flow. However, I am still curious whether aggregates form on the block and post in the absence of immobilized proteins (collagen/VWF) or in devices coated with Pluronics F-127? If so, what do the calcium dynamics, platelet accumulation, and forces look like? For example, in ref 22, platelets adhere, aggregate, and activate in the absence of immobilized proteins. How important are these immobilized proteins to the output of the device?

Reviewer #2 (Remarks to the Author):

The key concern on the original manuscript was that it was unclear why trauma-induced coagulopathy was the disease state selected for this study, since transfusion in trauma patients is primarily due to hyperfibrinolysis, consumptive coagulopathy/DIC and thrombocytopenia, rather than platelet dysfunction. More specifically, it was unclear what aspect of platelet dysfunction might be manifested in transfused trauma patients, and whether the detected platelet dysfunction (force) is confounded by consumptive coagulopathy, reduced von Willebrand factor levels and/or hemodilution.

There is insufficient information regarding pre-hospital fluid resuscitation and the timing of the blood test used in the force assay. The patients who subsequently went on to receive transfusion may have been more unwell or haemodynamically unstable, thus requiring larger fluid resuscitation. The reduced force may be related to haemodilution, and it is this which is predictive of subsequent transfusion. The authors therefore need to more adequately acknowledge the limitations of their clinical study and the interpretation of data from their Force assay.

The authors have partially addressed some of these important clinical issues in the Results section and they have cited previous studies showing platelet dysfunction in this patient group (Ref 43-45). They have re-plotted their data to confirm that Force reduction in trauma patients is not related to

thrombocytopenia or fibrinogen loss (Supp Fig 12-13). The authors were unable to assess von Willebrand factor levels due to difficulties in accessing trauma patients. However, they have now demonstrated an important influence of hemodilution on platelet force, even at hemodilution levels as low as 15% (Sup Fig 11). Thus, it is possible that additional confounding factors, in addition to platelet dysfunction, may contribute to the reduced platelet Force seen in trauma patients. This needs to be acknowledged by the authors.

Several other major concerns were raised in the original review.

1) It was unclear why the device dimensions were changed for the clinical study on trauma patients, and whether the changes impacted on the results and data interpretation;

The authors have now provided the reason (albeit not a clear rationale) as to why a 2nd generation device with larger dimensions was used for trauma patients. The authors have provided new Supp Fig 9 to show the design of 1st and 2nd generations of devices, also a comparable platelet area observed using 1st and 2nd Gen of devices (Supp Fig 9e-f). However, a much higher level of Force in 2nd Gen than 1st Gen was noted (Supp Fig 9g-h). Supp Fig 9 e-h is confusing, since the healthy blood was used for 1st Gen and trauma blood used for 2nd Gen (g,h). To address if the increased dimension enhances platelet contractile force, and the authors should validate the 2nd Gen device by assessing Force for healthy donors and compare with that readily obtained from trauma patients.

2) It is unclear how robust the data shown in newly provided supp Fig 11 which shows the impact of hemodilution on Force.

15% hemodilution seems to significantly impact the Force. However, this impact seemed highly variable, and did not reach statistical significance. This conclusion needs to be strengthened by performing additional studies on healthy donors with more sequential hemodilutions.

Minor points:

1. The Gen2 device has an aspect ratio of 25 μ m: 6.1 μ m for the small pillar. How do the authors achieve such a high aspect ratio (>4:1)? It would be useful to provide relevant information in the Methods section.

Reviewer #3 (Remarks to the Author):

I am satisfied with the improvements made to the manuscript by the authors.

November 2, 2018

Dear Editor and Reviewers:

Thank you for improving the quality of our revised manuscript. In our second revision, we have provided a new figure on the need for immobilized proteins (Supplemental Figure 3), conducted additional experiments on hemodilution (Supplemental Figure S12), edited the labels in Supplemental Figure S10, provide the amount of pre-hospital fluids received by trauma patients in Table S2, and made several changes to the manuscript. Edits to the text are highlighted in red underlined font. Below, we provide point-by-point responses to each of the reviewers.

Reviewer #1:

The authors have adequately addressed most of major comments in the revision and included several new experiments. There was one question from my previous review that was unanswered:

1. Fig. 3 addresses my previous comment regarding platelet activation caused by the device/flow. However, I am still curious whether aggregates form on the block and post in the absence of immobilized proteins (collagen/VWF) or in devices coated with Pluronic F-127? If so, what do the calcium dynamics, platelet accumulation, and forces look like? For example, in ref 22, platelets adhere, aggregate, and activate in the absence of immobilized proteins. How important are these immobilized proteins to the output of the device?

We have added a new supplemental figure with representative images from our experiments that addresses this reviewer's comment (Figure S4) and modified lines 195-198 in the manuscript. We find that platelet aggregates do not form on the block and post when they are coated with Pluronic F-127 to block soluble proteins from adsorbing onto their surface. Since the formation of aggregates is prevented, there is no accumulation, force, or calcium dynamics to be observed. Thus, we find that immobilized protein is critical to the adhesion of platelets onto PDMS.

The reviewer refers to reference 22 in the previous version of the manuscript, which is our *Lab on a Chip* paper in 2010 by Liang *et al.*, where we coated the tips of the microposts with fibronectin or fibrinogen. When we added a suspension of platelets in Tyrode buffer to a dish that contained arrays of microposts, the platelets settled to the bottom of the dish and attached to the tips of the microposts by adhesion to the immobilized ligand. The platelets were then activated by the addition of thrombin and allowed to aggregate for up to 1 hours. In this reference, the use of immobilize protein was essential to the formation of microaggregates.

**Figure S4.**

Reviewer 2:

The key concern on the original manuscript was that it was unclear why trauma-induced coagulopathy was the disease state selected for this study, since transfusion in trauma patients is primarily due to hyperfibrinolysis, consumptive coagulopathy/DIC and thrombocytopenia, rather than platelet dysfunction. More specifically, it was unclear what aspect of platelet dysfunction might be manifested in transfused trauma patients, and whether the detected platelet dysfunction (force) is confounded by consumptive coagulopathy, reduced von Willebrand factor levels and/or hemodilution.

We chose to examine platelet forces in the setting of trauma because, as this reviewer astutely claims, TIC is a mixed coagulopathy that manifests from many different pathological changes in coagulation and fibrinolysis, and is secondarily influenced by extrinsic factors such as hypothermia and hemodilution. However, there also exists evidence that platelets also quickly become dysfunctional during TIC, and that this dysfunction is strongly associated with clinically-important outcomes such as blood transfusion requirements and mortality (refs 43-45). Yet, there exist no good ways to rapidly isolate and measure platelet function to better understand its role in TIC and its influence on clinical trauma outcomes. There is also debate regarding why platelets may become dysfunctional. There is evidence that intrinsic changes in aggregation or cellular contraction may be present, and that extrinsic influences related to TIC, such as hemodilution, fibrinolysis, or consumptive coagulopathy may also contribute. Thus, our goals were the same as this reviewer's question – to improve the understanding of aspects of platelet dysfunction during TIC in addition to examining the potential clinical utility of the measurement for trauma medicine. Our *in vitro* experiments first confirmed the presence of decreased platelet forces in the simulated TIC model. This reviewer suggested a closer examination of hemodilution, which we have now reported in Figure S12.

The significant effect of hemodilution confirms a likely influence of hemodilution on platelet forces in our TIC model.

In our observational study, we then found that platelet forces were the only platelet-based measurement that could discriminate amongst Emergency Department trauma patients who required blood transfusions. Further analysis suggested by this reviewer regarding possible confounding effects of hemodilution has also confirmed that hemodilution, as measured by blood % hematocrit and the presence of prehospital fluid administration, was not a confounder in the clinical study. Platelet forces remained significantly and independently associated with blood transfusion ($p=0.01$) after adjusting for % hematocrit and for the presence of prehospital fluid administration in a multivariate logistic regression model. Hematocrit was also independently predictive of transfusion need in this multivariate model ($p=0.003$), which is likely an independent effect and is not surprising since many transfusion guidelines use hematocrit to guide transfusion decisions. However, hematocrit was clearly not a confounder because it was not directly associated with platelet forces (Pearson $R=0.5$, $p=0.59$) and it was not distributed unequally among the non-transfused and transfused trauma groups (Table S2). More, likely

Figure S12.

hematocrit acted as an effect modifier making transfusion more likely when both hematocrit and platelet force were decreased.

In summary, we found that the results from the TIC model and clinical study are in agreement that there exists intrinsic platelet dysfunction after trauma that is modified by hemodilution. This is an important addition to current knowledge regarding TIC and trauma medicine and supports the clinical utility of rapidly measuring platelet forces and attention to hemodilution in Emergency Department trauma patients. These results also illustrate how measurement of platelet forces can directly contribute to successfully understanding the role of platelets in complex coagulopathies like TIC. We have extensively revised the results section of the clinical study to incorporate these new analyses on lines 311-351, and thank the reviewer for the helpful suggestions that have strengthened this manuscript.

There is insufficient information regarding pre-hospital fluid resuscitation and the timing of the blood test used in the force assay. The patients who subsequently went on to receive transfusion may have been more unwell or haemodynamically unstable, thus requiring larger fluid resuscitation. The reduced force may be related to haemodilution, and it is this which is predictive of subsequent transfusion. The authors therefore need to more adequately acknowledge the limitations of their clinical study and the interpretation of data from their Force assay.

We now provide additional data on the pre-hospital fluids received by the trauma patients in Table S2. We find that hemodilution of the trauma patient was not associated with platelet forces and was not a confounder of the association between platelet forces and need for transfusions. Please see our response to the previous question for a detailed explanation.

We have also taken care to acknowledge study limitations in the discussion and to suggest further mechanistic and prospective clinical studies are required to confirm our results in lines 423-425 of the discussion.

The authors have partially addressed some of these important clinical issues in the Results section and they have cited previous studies showing platelet dysfunction in this patient group (Ref 43-45). They have re-plotted their data to confirm that Force reduction in trauma patients is not related to thrombocytopenia or fibrinogen loss (Supp Fig 12-13). The authors were unable to assess von Willebrand factor levels due to difficulties in accessing trauma patients. However, they have now demonstrated an important influence of hemodilution on platelet force, even at hemodilution levels as low as 15% (Sup Fig 11). Thus, it is possible that additional confounding factors, in addition to platelet dysfunction, may contribute to the reduced platelet Force seen in trauma patients. This needs to be acknowledged by the authors.

Again, we have demonstrated that the association between platelet forces and transfusion need is not confounded by hemodilution as measured by hematocrit and prehospital fluid administration. Please see our detailed response above. We do also agree that there are many possible factors may peripherally contribute to platelet dysfunction after trauma and have added this acknowledgement to the discussion as suggested on lines 423-425.

Several other major concerns were raised in the original review.

1) It was unclear why the device dimensions were changed for the clinical study on trauma patients, and whether the changes impacted on the results and data interpretation; The authors have now provided the reason (albeit not a clear rationale) as to why a 2nd generation device with larger dimensions was used for trauma patients. The authors have provided new Supp Fig 9 to show the design of 1st and 2nd generations of devices, also a comparable platelet area observed using 1st and 2nd Gen of devices (Supp Fig 9e-f). However, a much higher level of Force in 2nd Gen than 1st Gen was noted (Supp Fig 9g-h). Supp Fig 9 e-h is confusing, since the healthy blood was used for 1st Gen and trauma blood used for 2nd Gen (g,h). To address if the increased dimension enhances platelet contractile force, and the authors should validate the 2nd Gen device by assessing Force for healthy donors and compare with that readily obtained from trauma patients.

We acknowledge that the labels we used in this figure were confusing. We did in fact use healthy donors to validate the measurements we obtained with the second generation of the device. Regrettably, we had used "Gen 2 (Trauma)" as the label in the figure because that was the clinical application we intended for this version and how we referred to it within our group. In the revised manuscript, this figure is now Figure S10 and the labels for the data with control donors for both Gen 1 and Gen 2 have been corrected.

2) It is unclear how robust the data shown in newly provided supp Fig 11 which shows the impact of hemodilution on Force. 15% hemodilution seems to significantly impact the Force. However, this impact seemed highly variable, and did not reach statistical significance. This conclusion needs to be strengthened by performing additional studies on healthy donors with more sequential hemodilutions.

We thank the reviewer for this insight into improving our manuscript. We conducted new experiments with three additional donors and found that hemodilution causes a decrease in platelet forces and aggregate area that is statistical significance at 15% and 60% levels. We have added the new data to Figures S12 and modified lines 306-309 to state:

"To further investigate for an influence of hemodilution on platelet forces, we measured platelet forces in blood samples that were diluted with 0%, 15%, and 60% by volume with Ringer's solution and noted that forces were reduced with statistical significance (Fig. S12).."

Minor points:

1. The Gen2 device has an aspect ratio of 25 μm : 6.1 μm for the small pillar. How do the authors achieve such a high aspect ratio (>4:1)? It would be useful to provide relevant information in the Methods section.

We used SU-8 photoresist to fabricate high aspect ratio structures on the silicon master. We have modified lines 76-77 in the manuscript to provide this information. In addition, the details of the process are contained in the caption for Figure S1.

Reviewer 3:

I am satisfied with the improvements made to the manuscript by the authors.

In closing, we hope that the improvements in our manuscript meet your satisfaction. We thank you for considering our manuscript and look forward to a favorable outcome. On behalf of my fellow co-authors, we appreciated the valuable time of the reviewers and careful attention they have put into this process.

Sincerely,

Nathan J. Sniadecki, Ph.D.

REVIEWERS' COMMENTS:

Reviewer #1 (Remarks to the Author):

The authors have adequately addressed all of my comments.

Reviewer #2 (Remarks to the Author):

The authors have done a good job addressing the residual concerns raised in previous reviews. I am satisfied with the changes made to the manuscript.

Shaun Jackson